# Noise Stability of Transformer Models

**Themistoklis Haris**[*]
Department of Computer Science
Boston University
Boston, USA
`tharis@bu.edu`

**Zihan Zhang**
National Institute of Informatics
Tokyo, Japan
`zihan@nii.ac.jp`

**Yuichi Yoshida**
National Institute of Informatics
Tokyo, Japan
`yyoshida@nii.ac.jp`

## Abstract

Understanding simplicity biases in deep learning offers a promising path toward developing reliable AI. A common metric for this, inspired by Boolean function analysis, is average sensitivity, which captures a model's robustness to single-token perturbations. We argue that average sensitivity has two key limitations: it lacks a natural generalization to real-valued domains and fails to explain the "junta-like" input dependence we empirically observe in modern LLMs. To address these limitations, we propose *noise stability* as a more comprehensive simplicity metric. Noise stability expresses a model's robustness to correlated noise applied to *all* input coordinates simultaneously. We provide a theoretical analysis of noise stability for single-layer attention and ReLU MLP layers and tackle the multi-layer propagation problem with a covariance interval propagation approach. Building on this theory, we develop a practical *noise stability regularization* method. Experiments on algorithmic and next-token-prediction tasks show that our regularizer consistently catalyzes grokking and accelerates training by approximately 35% and 75% respectively. Our results sculpt a new connection between signal propagation in neural networks and interpretability, with noise stability emerging as a powerful tool for understanding and improving modern Transformers.

## 1 Introduction

Simplicity Biases have been a promising direction of study in recent years (Shah et al., 2020; Vasudeva et al., 2024; Bhattamishra et al., 2022) as they provide a unifying framework for generalization, interpretability and robustness. Neural networks, including Large Language Models (LLMs), often converge to the simplest possible functions that explain the training data. Because simpler functions are inherently more interpretable and robust, this bias provides a solid theoretical framework for improving model reliability.

To quantify simplicity, current research often draws on concepts from Boolean function analysis (O'Donnell, 2021). In particular, theoretical work on Transformers (Vaswani et al., 2017) has highlighted average sensitivity, the expected change in a model's output given a single-token perturbation, as a key metric. Bhattamishra et al. (2022) formally studied this metric in Transformers, showing that they learn functions with lower sensitivity than LSTMs. Subsequent work has linked average sensitivity to learnability: Hahn (2020) and Hahn & Rofin (2024) demonstrated that Transformers struggle to learn functions with high sensitivity, such as Parity. Further validating its utility, Vasudeva et al. (2024) connected average sensitivity to the *grokking* phenomenon and proposed an extension of the metric beyond the Boolean domain.

---

[*]This work was conducted while the author was an intern at NII.

Despite its usefulness, average sensitivity has notable drawbacks. Theoretically, its origins in Boolean analysis do not readily extend to real-valued domains. Empirically, it fails to fully explain the "junta-like" input dependence we observe in models like GPT-2, GEMMA, and ROBERTA, where outputs rely on a small subset of inputs.

To address these shortcomings, we propose to instead quantify simplicity bias via *noise stability*. Unlike average sensitivity's one-by-one perturbations, noise stability measures a function's resilience to noise applied to *all inputs simultaneously*, offering more robust spectral concentration guarantees. This approach naturally extends to real-valued domains via the Ornstein-Uhlenbeck operator in the Gaussian measure, preserving a formal connection to the function spectrum and enabling a more powerful theoretical analysis.

## 1.1 OUR CONTRIBUTIONS

Our primary contributions are the following:

1. We observe that LLMs like GPT-2 exhibit a "junta-like" input dependence (Figure 1), a phenomenon not fully captured by average sensitivity and its extensions (Section 3). To better characterize this behavior, we propose **noise stability** (Section 4) as a comprehensive simplicity metric that naturally extends to real-valued domains.

2. We derive theoretical results on noise stability for single-layer Transformers and ReLU FFNs. We also provide novel insights for the multi-layer setting (Section 5) through a proxy recurrence-based analysis and a stability interval propagation technique.

3. We propose **noise stability regularization** (Section 6), a method that consistently accelerates grokking across synthetic (modular addition, sparse parity) and non-synthetic (next-token-prediction) benchmarks, reducing the training time required for generalization by $\approx 35\%$ and $75\%$ respectively.

## 1.2 RELATED WORK

**Simplicity Bias in Deep Learning.** The tendency of neural networks to converge to simple functions has been a subject of intense recent study. This simplicity bias (SB) is analyzed from several perspectives. One line of research connects SB to spectral concentration in the Conjugate or Neural Tangent Kernel of networks (Yang & Salman, 2019; Emami et al., 2021; Vasudeva et al., 2024). Another uses Fourier analysis to characterize SB as a bias toward low-frequency functions (Xu et al., 2019; Rahaman et al., 2019). A large body of work investigates SB through the lens of training dynamics, often in shallow or linear networks (Morwani et al., 2023; Zhang et al., 2019; Yun et al., 2020; Chen et al., 2020; Boursier & Flammarion, 2024; Chizat & Bach, 2020; Gatmiry et al., 2024; Tsoy & Konstantinov, 2024). Beyond its mechanisms, SB has been linked to generalization (Valle-Perez et al., 2018), though it can sometimes lead to degenerate solutions (Shah et al., 2020), and has been correlated with adversarial robustness (Min & Vidal, 2024; Chen et al., 2020). Specific to our focus, recent work has begun to explore SB in Transformers, particularly through the lens of token-to-token interactions in shallow models (Teney et al., 2025; Rende et al., 2024).

**Sensitivity Analysis in Transformers.** To develop a computationally tractable proxy for spectral concentration, recent work has adopted *average sensitivity* from Boolean function analysis. Bhattamishra et al. (2022) showed that Transformers are more biased towards low-sensitivity functions than LSTMs, enabling generalization even with noisy labels. Hahn (2020); Hahn & Rofin (2024) established that Transformers struggle to learn high-sensitivity functions like parity, despite having the capacity to represent them. Further, Vasudeva et al. (2024) demonstrated that average sensitivity can also serve as a metric for tracking progress in grokking.

**Signal Propagation in Neural Architectures** Motivated by the challenge of gradient instability during training (e.g., vanishing or exploding gradients), a rich body of literature has established a theory of signal propagation in neural networks (Poole et al., 2016; Ji et al., 2023; Lee et al., 2019; Kohan et al., 2023; Lou et al., 2022; Wang et al., 2024). Leveraging tools from statistical physics, mean-field dynamical systems (Schoenholz et al., 2016), kernel methods (Martens et al., 2021), and random matrix theory (Pennington et al., 2017), this now-mature theory offers critical insights

into maintaining models at the "edge of chaos"—a state of criticality where signal information is preserved without diverging or collapsing. Although theoretical analyses are frequently constrained to models at initialization, they provide valuable guidelines regarding the efficacy of optimization techniques such as dropout (Pretorius et al., 2018) and residual connections (Fischer et al., 2023), the mechanics of which were previously under-theorized.

Central to this literature are two key quantities: the correlation between the pre-activations of distinct inputs, and the pre-activation variance[1]. The notion of noise stability proposed in our work can be viewed as a more parsimonious analogue to signal correlation and $C$-maps. Indeed, our results on propagating noise stability through MLP and Attention layers mirror similar findings in the signal propagation literature. However, we instead arrived at noise stability through questions of spectral concentration and simplicity bias, rather than initialization and training stability. This distinction suggests a compelling link between the two fields of study that can be further explored in future work.

Signal propagation has also been examined within Transformer architectures to address pathological issues such as unstable gradients and rank collapse (Saada et al., 2024; Noci et al., 2022; Huang et al., 2023). These works also generally adopt a dynamical systems perspective; in contrast, we utilize a discrete viewpoint and explicitly draw connections to simplicity biases. Unifying these theoretical landscapes to encompass both discrete and continuous domains remains a promising direction for future research.

Finally, noise-related regularization is not unprecedented. Hua et al. (2023), for instance, propose a noise-stability regularization method for fine-tuning Transformers. Our approach differs fundamentally in motivation, scope of application, and the definition of correlated noise.

**Generalizing Sensitivity to Continuous Domains.** The concept of average sensitivity has been generalized to real-valued domains via *geometric influences* (Keller et al., 2012; 2014). This formulation is equipped with an analogue of Friedgut's junta theorem for continuous spaces (Bouyrie, 2017), unifying prior results across various discrete and continuous measures (Benjamini et al., 2016; Wimmer, 2014).

**Noise Stability and Sensitivity.** Our work is most closely related to that of Li & Mossel (2025), who study *noise sensitivity*—a dual measure to our noise stability—for hierarchical functions. They use an inductive argument to propagate sensitivity bounds across layers in a simple, non-intersecting neural network. Though they do not study Transformers in practice, their layer-wise propagation strategy directly inspired our approach for noise stability intervals in multi-layer Transformers.

## 2 Setup

We define a simplified *Transformer* as an $L$-layer model that maps an input sequence $X \in \mathbb{R}^{n \times d}$ to a distribution over $n_c$ classes. Each layer $i \in [L]$ contains $H$ attention heads. A head $j$ takes the layer input $Y_i \in \mathbb{R}^{n \times d}$ and computes an output $a_{i,j} \in \mathbb{R}^{n \times d}$ via:

$$a_{i,j} = \sigma(Y_i^T W_{Q,i,j}^T W_{K,i,j} Y_i)(Y_i^T W_{V,i,j})$$

Here, $W_{Q,i,j}, W_{K,i,j}, W_{V,i,j} \in \mathbb{R}^{d \times d}$ are weight matrices and $\sigma$ is the row-wise softmax[2]. The head outputs are concatenated and passed through a Multi-Layer Perceptron (MLP) with a ReLU activation, $\phi(x) = \max\{0, x\}$: $\hat{a}_i = \phi((a_{i,1} \circ \cdots \circ a_{i,H}) W_\phi^{(i)})$. The final layer's output $\hat{a}_L$ is then mean-pooled and projected to produce class logits $z \in \mathbb{R}^{n_c}$ using an output matrix $W_O \in \mathbb{R}^{d \times n_c}$.

For theoretical simplicity, this definition omits elements like residual connections, layer normalization, and attention masks, though we include them in our experiments.

---

[1]Poole et al. (2016) explicitly define iterative maps for these quantities: $Q$-maps capture variance and $C$-maps capture correlation.

[2]For convenience, in our theoretical results we will ignore the $1/\sqrt{d}$ factor that appears in these expressions.

## 2.1 BOOLEAN FUNCTION ANALYSIS

Our work draws on Boolean function analysis, which studies functions $f : \{\pm 1\}^n \to \mathbb{R}$ by expanding them as multilinear polynomials via the *Fourier spectrum*: $f = \sum_{U \subseteq [n]} \hat{f}_U \chi_U$, where $\chi_U(x) := \prod_{i \in U} x_i$ are the basis functions. For a full overview, see Appendix A.

A key property is the *influence* of a coordinate $i \in [n]$, which measures the expected impact of flipping the input $x_i$:

$$\text{Inf}_i[f] := \mathop{\mathbb{E}}_{x \sim \{\pm 1\}^n} \left[ \left( \frac{f(x) - f(x^{\oplus i})}{2} \right)^2 \right] = \sum_{S \ni i} \hat{f}_S^2 \tag{1}$$

The *total influence* across all coordinates is the **average sensitivity**, $I[f] = \sum_{i=1}^n \text{Inf}_i[f]$, a common measure of a function's robustness to noise.

## 3 MODELS ARE OFTEN "SIMPLER" THAN EXPECTED

The "simplicity" of a Boolean function $f : \{\pm 1\}^n \to \mathbb{R}$ is formalized by its dependence on a few variables. One way to characterize simplicity is *spectral concentration*, where the most of the Fourier mass is on low-degree coefficients. A function is $(\varepsilon, k)$-spectrally concentrated if its mass on terms of degree $k$ or higher is bounded:

$$\sum_{j=k}^n W^j[f] \leq \varepsilon \cdot ||f||_2^2, \quad \text{where} \quad W^j[f] := \sum_{|U|=j} \hat{f}_U^2$$

A stricter notion is a *k-junta*, a function that depends on at most $k$ coordinates. A function with low average sensitivity is simple in both senses: a function is always $(\varepsilon, I[f]/\varepsilon)$-spectrally concentrated, and *Friedgut's Junta Theorem* (Kelman et al., 2021; Friedgut, 1998) shows it must also be structurally close to a junta:

**Theorem 3.1.** *For every $\epsilon > 0$, there exists a $k$-junta $g : \{-1, 1\}^n \to \{-1, 1\}$ such that $\mathbb{P}[f(x) \neq g(x)] \leq \epsilon$, where the number of variables $k$ on which $g$ depends is bounded by $k \leq 2^{O(I(f)/\epsilon)}$.*

While prior work has used average sensitivity to analyze model simplicity (Vasudeva et al., 2024; Hahn & Rofin, 2024), we argue it has significant theoretical and empirical drawbacks as a metric for simplicity bias.

**Theoretical Drawbacks: Extending to Real-Valued Domains** First, the theoretical foundation of average sensitivity in Boolean domains is difficult to extend to the real-valued functions of deep learning. Approaches using generalized domains that mimic finite fields[3] are cumbersome, as sensitivity is not naturally defined, and its estimation via sampling is impractical.

A more robust alternative is **geometric influence** (Keller et al., 2012), defined for a smooth function $f : \mathbb{R}^n \to \mathbb{R}$ and measure $\mu$ as:

$$I^{\mathcal{G}}[f] := \sum_{i=1}^n ||\partial_i f||_{L^1(\mu)}$$

Crucially, this measure has strong theoretical backing, including an analogue of Friedgut's junta theorem for hypercontractive measures (Bouyrie, 2017). This makes it a more suitable tool for analyzing neural networks and we use it in our empirical study below.

**Empirical Drawbacks: Mismatch with LLM Behavior** Second, average sensitivity fails to capture the empirical simplicity of LLMs. The exponential bound on influential variables from Friedgut's theorem is too loose to explain the behavior of modern Transformers, which exhibit far stronger influence concentration. To demonstrate this, we analyze the geometric influence of each input token for GEMMA-2B, ROBERTA, and GPT-2 on sequences of $n = 256$ tokens. Friegut's theorem predicts $\leq 1024$ variables with influence at least $0.1 \cdot I[f]$, yet Figure 1 only shows 5-10 such variables exist, meaning the bound is loose. Our analysis also reveals three key patterns with further experimental details available in Appendix B:

---

[3] See (O'Donnell, 2021), Chapter 8, for an exposition on Boolean Function Analysis on the hypergrid.

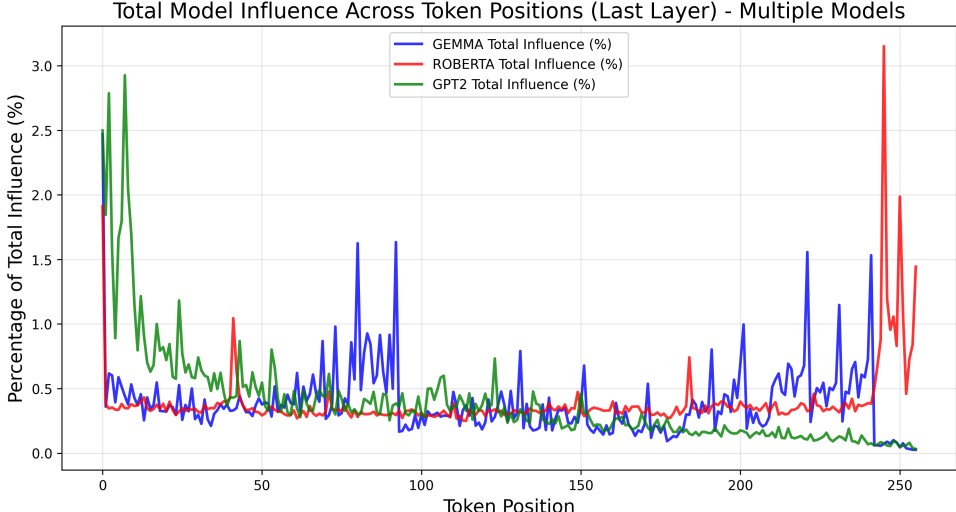

Figure 1: Comparing the per-coordinate geometric influence of three models for $n = 256$.

- **Junta-like Concentration:** A small subset of tokens have disproportionally high influence.

- **Positional Bias:** The first and last tokens are consistently the most influential. This is part agrees with observations made in the KV Cache Compression literature about "attention sinks" (Xiao et al., 2023).

- **Sensitivity:** Every token has a non-zero influence, indicating that the models are sensitive to all inputs, even if asymmetrically.

## 4 NOISE STABILITY AS A MEASURE OF CONCENTRATION

For a more holistic characterization of simplicity that offers finer control over spectral concentration and easily generalizes to real-valued domains, we propose **noise stability**, a concept from Boolean Function Analysis (O'Donnell, 2021) that measures a function's resilience to correlated noise *applied to all inputs simultaneously*.

This concept extends naturally to real-valued functions in $L^2(\gamma)$ by leveraging the Ornstein-Uhlenbeck (OU) semigroup $T_\rho$. This framework uses the basis of Hermite polynomials under the standard Gaussian measure $\gamma \equiv \mathcal{N}(0, 1)$, allowing for a direct spectral interpretation[4]. The correlated pair $(X, Y)$ is generated by adding scaled Gaussian noise to $X$:

**Definition 4.1.** Let $f \in L^2(\gamma)$ where $\gamma$ is the Gaussian measure on $\mathbb{R}^n$. For $\rho \in (0, 1)$, let $X \sim \gamma$ and let $Y = \rho X + Z\sqrt{1 - \rho^2}$, where $Z \sim \gamma$ is independent of $X$. The noise stability of $f$ is:

$$\text{Stab}_\rho(f) := \mathbb{E}_{(X,Y)}[f(X)f(Y)] \tag{2}$$

Noise stability is useful because it relates directly to the function's spectrum through its Hermite-Fourier coefficients $\tilde{f}(\alpha)$, as shown in Appendix C:

$$\text{Stab}_\rho(f) = \sum_{\alpha \in \mathbb{N}^d} \rho^{|\alpha|} \tilde{f}(\alpha)^2 \tag{3}$$

This connection allows us to formally bound a function's spectral concentration. The following lemma shows that if a function is highly stable (i.e., its stability is close to its total variance), its Fourier mass must be concentrated on low-degree coefficients. For a fixed correlation $\rho$ and spectral tail budget $\varepsilon$, the degree of concentration $T$ becomes smaller as the ratio $\delta/\varepsilon$ approaches zero.

---

[4]See Appendix C, following Andersson & Sjögren (2012), for an introduction to OU Semigroup Theory.

**Lemma 1** (Spectral Concentration via Stability). *Let $f \in L^2(\gamma^n)$. If $Stab_\rho(f) \geq (1-\delta)\|f\|_2^2$ for some $\rho \in (0,1)$ and $0 < \delta < \varepsilon < 1$, then $f$ is $(\varepsilon, T)$-spectrally concentrated for any*

$$T \geq \log_{\frac{1}{\rho}}\left(1 - \frac{\delta}{\varepsilon}\right)$$

*Proof Sketch.* The proof follows from analyzing the action of the Ornstein-Uhlenbeck semigroup $T_\rho$ on the Hermite expansion of $f$. The full proof is in Appendix D. $\qquad\square$

**Spectral Concentration Bounds: Sensitivity vs. Stability** We compare the predicted Fourier tail mass (percentage of weight in degrees $\geq 15$ with $n = 256$) for several Transformer models, using bounds derived from average sensitivity versus those from noise stability. The results in Table 1 show that the noise stability bound offers a more accurate estimate of spectral concentration.

| Model | Tail Mass Bound from $I[f]$ | Tail Mass Bound from $\text{Stab}_\rho(f)$ |
|---|---|---|
| GPT-2 | 0.003 | **0.0005** |
| BERT | 0.04 | **0.02** |
| ROBERTA | 0.19 | **0.02** |
| GEMMA | 0.043 | **0.0157** |

Table 1: Predicted Fourier tail mass (percentage of weight in degrees $\geq 15$) for Transformer models.

## 5 ANALYSIS OF NOISE STABILITY IN TRANSFORMER MODELS

We now present our theoretical results on the noise stability of Transformer components. We begin by analyzing a single ReLU MLP layer and an attention layer, and then use these results to analyze the propagation of stability through a multi-layer network.

### 5.1 NOISE STABILITY OF A SINGLE ReLU MLP LAYER

We first analyze the stability of an MLP layer with a ReLU activation, a result closely related to the arc-cosine kernel. Consider a pair of $\rho$-correlated standard Gaussian inputs $(X, Y)$, whose joint distribution is:

$$(X, Y) \sim \mathcal{N}\left(0, \begin{pmatrix} 1 & \rho \\ \rho & 1 \end{pmatrix}\right)$$

The noise stability of the ReLU function is given by the following theorem.

**Theorem 5.1.** *The noise stability of the ReLU function under the standard Gaussian measure is:*

$$\mathbb{E}_{(X,Y)}[ReLU(X)ReLU(Y)] = \frac{1}{2\pi}\left(\sqrt{1-\rho^2} + \rho(\pi - \arccos\rho)\right)$$

*Proof.* The proof is by direct integration and can be found in Appendix E. $\qquad\square$

While exact, this expression is unwieldy for analyzing layer composition. For practical purposes, it is well-approximated by its second-order Taylor expansion around $\rho = 0$:

$$\frac{1}{2\pi}\left(\sqrt{1-\rho^2} + \rho(\pi - \arccos\rho)\right) \approx \frac{1}{2\pi} + \frac{1}{4}\rho + \frac{1}{4\pi}\rho^2 + O(\rho^3). \tag{4}$$

### 5.2 NOISE STABILITY OF A SINGLE ATTENTION LAYER

We next analyze the noise stability of a single attention layer, defined as $f(X) = \sigma(XW_Q W_K^T X^T) \cdot XW_V$. The analysis depends critically on the structure of the product $W := W_Q W_K^T$, so we consider three representative cases.

**The Identity Case** ($W = I_d$)   When $W$ is the identity matrix, the attention mechanism simplifies. In the high-dimensional limit, the attention matrix $\sigma(XX^T)$ converges to the identity matrix $I_n$, causing the layer to act as a linear transformation. This results in a linear relationship between input and output stability (see Figure 2).

**Theorem 5.2.** *Let $X \sim \mathcal{N}(0, I_{n \times d})$ and $Y = \rho X + Z\sqrt{1 - \rho^2}$ for $Z \sim \mathcal{N}(0, I_{n \times d})$ independent of $X$. Let $f(X) = \sigma(XX^T)XW_V$. Then for any $i \in [n], j \in [d]$:*

$$\lim_{d \to \infty} \mathbb{E}[f(X)_{ij} f(Y)_{ij}] = \rho \cdot ||(W_V)_{:,j}||_2^2 + o(1)$$

*Proof Sketch.* As $d \to \infty$, we show that $\sigma(XX^T) \to I_n$ in probability. The stability calculation then reduces to that of a linear layer, with a cost of $o(1)$. The full proof is in Appendix F.2.   $\square$

**The Low-Rank Case** ($W = UU^T$)   If $W$ is a random low-rank matrix, where $U \in \mathbb{R}^{d \times k}$ with $k \ll d$, the analysis reduces to the identity case. The matrix $U$ acts as a Johnson-Lindenstrauss transform, projecting the input row vectors into a $k$-dimensional space while approximately preserving their inner products (Matoušek, 2008). Consequently, the projected attention matrix again converges to the identity, and the stability remains linear in $\rho$.

**The Unstructured Case** ($W \sim \mathcal{N}(0, I_{d \times d})$)   When $W$ is a random Gaussian matrix, modeling a randomly initialized layer, the behavior changes. For large $d$, we the attention matrix $A_X = \sigma(XWX^T)$ concentrates towards a random permutation matrix, meaning each output token attends to a single, randomly chosen input token.

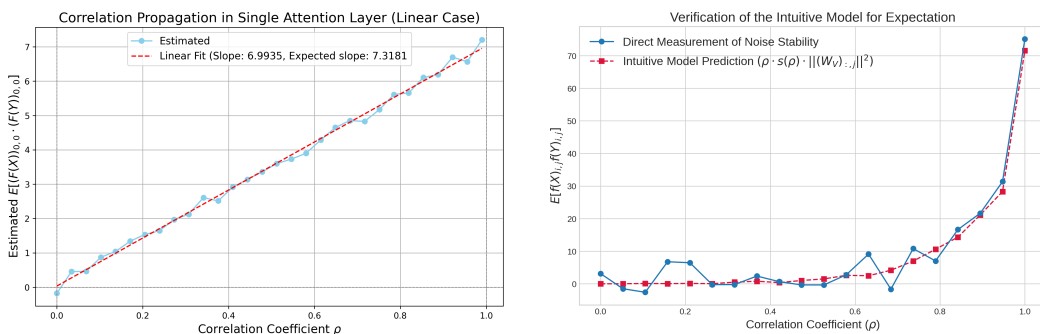

Figure 2: Stability of Single Layer Attention (Identity and Unstructured)

The stability now depends on the consistency of this permutation. For a given output row $i$, suppose $A_X$ selects input row $k$ and $A_Y$ selects input row $k'$. The stability is non-zero only if the attention pattern is preserved ($k = k'$). Let $s(\rho) := \mathbb{P}(k = k')$ be the probability that the attention pattern is stable (see Figure 3 for an illustration). The total stability is the product of the linear stability and this probability factor:

$$\mathbb{E}[f(X)_{ij} f(Y)_{ij}] = \begin{cases} \rho \cdot ||(W_V)_{:,j}||_2^2 & \text{if } k = k' \\ 0 & \text{if } k \neq k' \end{cases}$$

Averaging over the randomness of the patterns we obtain the following result, which we verify empirically in Figure 2 (see Appendix G for the proof).

**Theorem 5.3.** *Let $X \sim \mathcal{N}(0, I_{n \times d})$ and $Y = \rho X + Z\sqrt{1 - \rho^2}$ for $Z \sim \mathcal{N}(0, I_{n \times d})$ independent of $X$. Let $f(X) = \sigma(XX^T)XW_V$. Then for any $i \in [n], j \in [d]$, we have:*

$$\lim_{d \to \infty} \mathbb{E}[f(X)_{ij} f(Y)_{ij}] \overset{p}{=} \rho \cdot s(\rho) \cdot ||(W_V)_{:,j}||_2^2 + o(1), \text{ with } s(\rho) = n \int_{\mathbb{R}^2} \Phi_{\rho^2}(x,y)^{n-1} f_{\rho^2}(x,y)$$

*where $\Phi_c, f_c$ are the joint CDF and PDF of a bivariate normal distribution with correlation c.*

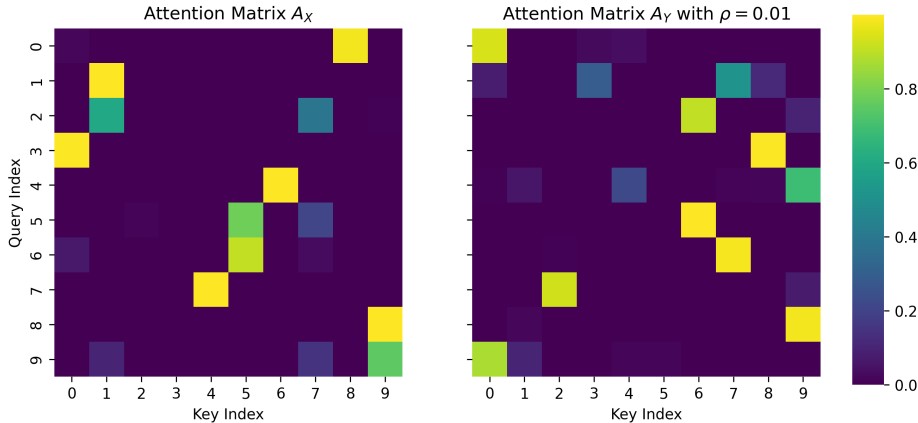

Figure 3: Comparing $A_X$ and $A_Y$ for $d = 128$ and $\rho = 0.01$.

## 5.3 STABILITY PROPAGATION IN DEEP TRANSFORMERS

In the single-layer setting, we have shown that ReLU MLPs dampen stability (Theorem 5.1), while attention layers can preserve it (Theorem 5.2). This raises the question whether such an analysis can be performed for the multi-layer setting as well.

**FFN Stability Propagation as a Recurrence and Weak Dampening**  Consider a ReLU Feed-Forward Network (FFN) and suppose we ignore inter-layer distributional shifts. Such a thought experiment is not without precedent in the moment propagation literature for neural networks (Wright et al., 2024). In this model, the correlation $\rho_L$ after layer $L$ follows the recurrence:

$$\rho_L = \frac{1}{2\pi} \left( \sqrt{1 - \rho_{L-1}^2} + \rho_{L-1}(\pi - \arccos(\rho_{L-1})) \right) \tag{5}$$

Solving this non-linear recurrence analytically is difficult, so we shall instead use the linear approximation from Equation (4), to get the following proxy recurrence, which can be solved more easily:

$$\rho_L = \frac{1}{2\pi} + \frac{1}{4}\rho_{L-1} \implies \rho_L = \frac{2}{3\pi} + \left( \frac{1}{2} - \frac{2}{3\pi} \right) \cdot \left( \frac{1}{4} \right)^{L-1}$$

This suggests that noise stability for multi-layer ReLU FFNs exhibits *weak dampening*, converging to the fixed point $\frac{2}{3\pi}$. This is confirmed by a numerical evaluation of Equation (5) in Figure 4b. Indeed, Figure 4a shows that a multi-layer MLP does exhibit weak dampening.

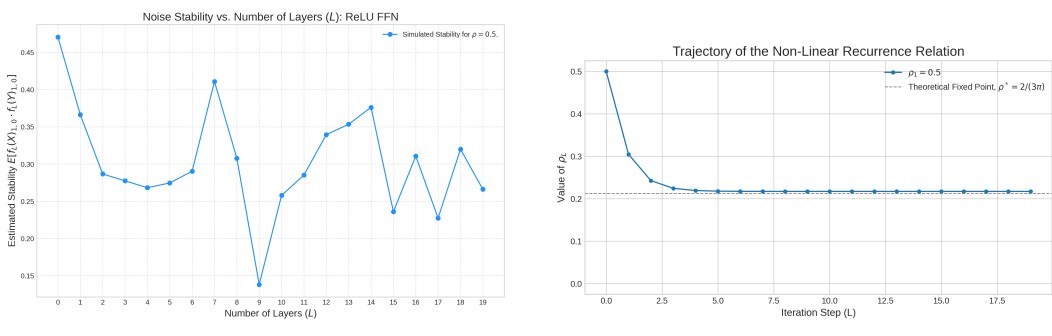

(a) Stability of a deep MLP network.  (b) Numerical solution to Equation (5)

For multi-layer Transformers, a recurrence relation analysis does not yield weak dampening. If we let $W_Q W_K^T = I$ and $||(W_V)_{:,j}||_2 = \gamma \leq 1$, a recurrence similar to Equation (5) would yield candidate fixed points of $\frac{2}{\pi(4-\gamma^2)}$. However, we observe that for $\gamma < 1$ the noise stability signal actually dampens fully to zero, suggesting that the attention map alters the distribution enough to preclude the weak dampening behavior. For more details, see Appendix H.

**Noise Stability Intervals**  A more formal approach is to track rigorous upper and lower bounds on the noise stability as they propagate through the network. We derive such bounds for individual MLP and attention layers (Appendix I.1 and Appendix I.2). Further empirical work is needed to determine the tightness of these bounds in practice.

## 6  NOISE STABILITY REGULARIZATION AND ITS BENEFITS

We established in Section 4 and Section 5 that high noise stability is a desirable property for creating robust and interpretable models. To this end, we introduce a regularizer designed to orient a model's training towards to or away from stable functions. We designed our regularizer to be (1) **differentiable** with respect to the model's parameters, and (2) **data-dependent**, meaning that the regularization should be evaluated on the model's outputs for training data, not just its parameters.

**Definition 6.1** (Noise Stability Regularization). Let $M : [U]^N \to \Delta(C)$ be a model, $X \in [U]^N$ be an input sequence, $S \in \{0, 1\}$ be the orientation parameter, and $\rho \in [-1, 1]$ be a correlation parameter. The $S$-**oriented noise stability regularizer** is defined as:

$$R_{M,S,\rho}(X) = (-1)^S \cdot \sum_{i=1}^{C} M(X)_i \cdot M(Y)_i, \text{ where:} \tag{6}$$

$$Y_i = \begin{cases} X_i, & \text{with probability } \frac{1+\rho}{2} \\ Z \sim \text{uniform}([U]), & \text{otherwise.} \end{cases} \tag{7}$$

Setting the orientation parameter $S = 1$ *encourages stability*. For a differentiable loss function $\ell$, the regularized loss then becomes $\ell_{\text{reg}}(M, X) := \ell(M, X) + \gamma \cdot R_{M,S,\rho}(X)$ where $\gamma$ is a hyperparameter controlling the regularization strength. We test the effect of positively-oriented noise stability ($S = 1$) by training a Transformer from scratch on two tasks known to exhibit "grokking": *noisy k-sparse parity* and *modular addition*.

Note that calculating the noise stability regularizer requires just one additional forward pass through the model per training iteration. It is an interesting direction whether the regularization can be applied in a more cost-effective manner.

**Experimental Setup**  We use a standard decoder-only Transformer (Appendix J). For all experiments, we compare regularized and non-regularized models. All other hyperparameters, including the random seed, are held constant across runs (Figure 5).

NOISY $k$-SPARSE PARITY (NSP)  We learn the function $f(x) = \bigoplus_{i \in I} x_i$ for an input $x \in \{0, 1\}^n$ and a secret sparse index set $I \subset [n]$ of size $k$. Each training label is flipped with a fixed probability $\eta$. This problem is well-studied in learning theory (Chen et al., 2024; Feldman et al., 2009), and Transformers can solve it for small values of $k$ (Bhattamishra et al., 2022). We test on inputs of length $n \in [10, 100]$ with $k \in \{2, 3\}$, using $(\gamma, \rho) = (0.05, 0.05)$.

MODULAR ADDITION  This task is to compute $(X + Y) \pmod{K}$. We study it as a standard benchmark for grokking in Transformers (Nanda et al., 2023). For our experiments, we use a prime modulus $K = 113$, train for $10,000$ iterations, and set $(\gamma, \rho) = (0.75, 0.25)$.

**Noise Stability Regularization Catalyzes Grokking.**  Training Transformers on these tasks exhibits an "emergence" phenomenon, where validation loss drops suddenly after a long period of stagnation. We find that noise stability regularization acts as a catalyst for this emergence. For modular addition, regularization reduces the iterations required from $\approx 4500$ to $\approx 3300$, a $36\%$ acceleration. We observe a similar $\approx 35\%$ speed-up for the noisy sparse parity task (Figure 5).

**Noise Stability Evolution During Training**  By observing the noisy sparse parity task, we find that a Transformer's noise stability naturally decreases during training to match the target function, serving as a leading indicator of generalization (Appendix J.2).

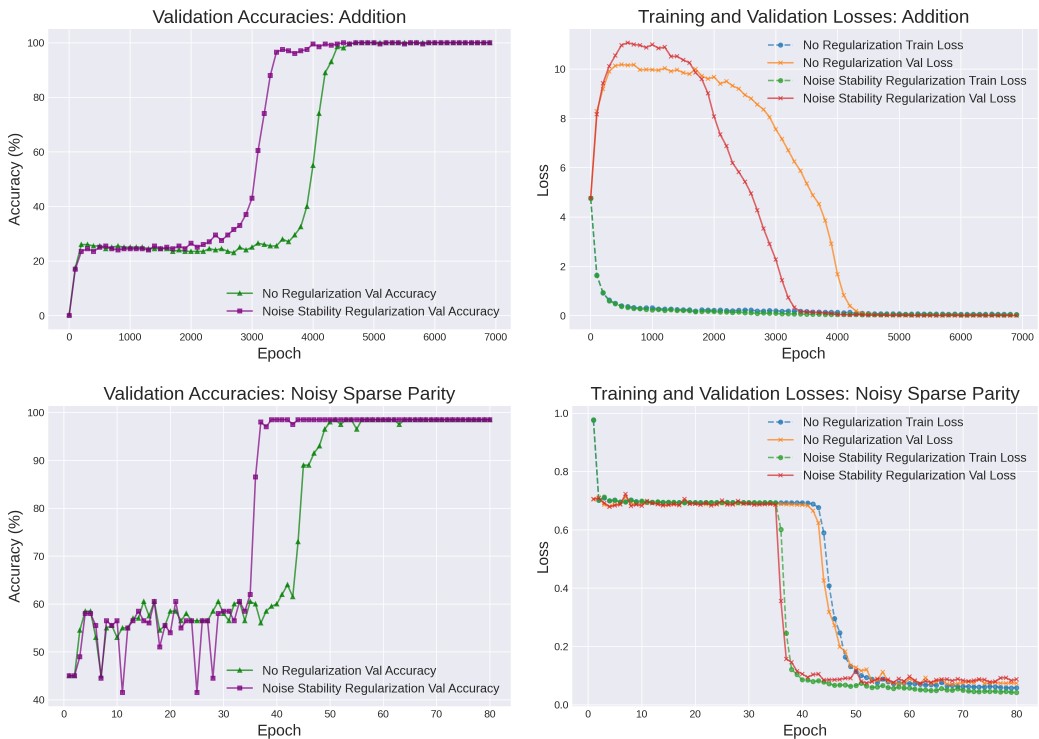

Figure 5: Noise Stability Regularization accelerates training.

## 6.1 NON-SYNTHETIC EXPERIMENTS ON LANGUAGE GENERATION

We also tested noise stability regularization for the task of next-token-prediction on *WikiText-2-v1* (Appendix J.3). We observed that regularized training reaches high validation accuracy in $\approx 75\%$ fewer iterations (Figure 6). The noise stability of the regularized model notably stays high, while non-regularized models become increasingly unstable.

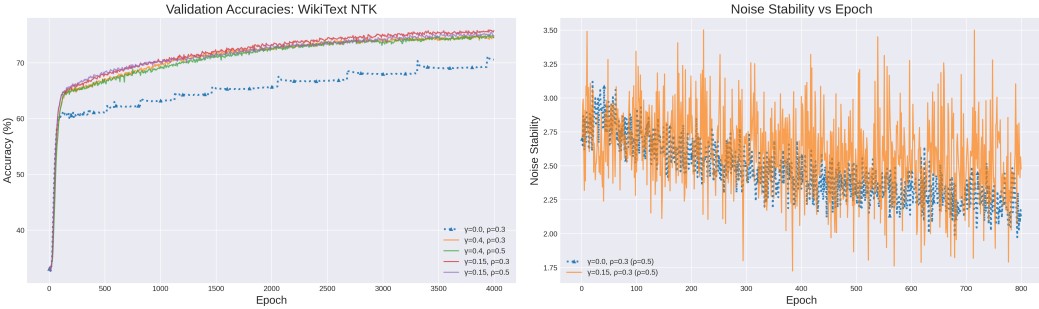

Figure 6: Noise Stability Regularization for Next-Token-Prediction (NTK) on WikiText-2

## 7 CONCLUSION

In this work, we introduced *noise stability* as a measure of simplicity bias in Transformers, arguing theoretically and empirically that it better explains the spectral concentration observed in LLMs than average sensitivity. We also proposed a noise stability regularizer and found that it unexpectedly catalyzes grokking, offering a potential avenue to accelerate model training. Our findings open several avenues for future research, including demystifying the mechanism of moment propagation in deep networks and understanding the practical limits and theoretical underpinnings of noise stability regularization in LLM settings.

ACKNOWLEDGEMENTS

We thank the anonymous ICLR 2026 Area Chair (QcA8) for highlighting the extensive literature regarding signal propagation. TH additionally thanks the faculty and staff at the National Institute of Informatics for providing a stimulating research environment throughout the summer internship program. YY is supported by JSPS KAKENHI Grant Number 24K02903.

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

APPENDIX CONTENTS

# A   ESSENTIALS OF BOOLEAN FUNCTION ANALYSIS

We now review some basic facts on Boolean Function Analysis. A Boolean function is defined on the hypercube: $f : \{\pm 1\}^n \to \mathbb{R}$. Therefore, we can think of the space of boolean functions as $\mathbb{R}^{2^n}$. It is a fundamental fact that every boolean function can be represented *uniquely* as a multilinear polynomial. This is a natural outcome of considering the following set of vectors in $\mathbb{R}^{2^n}$:

$$B = \left\{ \chi_U(x) = \prod_{i \in U} x_i : U \subseteq [n] \right\}$$

For any $U, V$ we have that:

$$\langle \chi_U, \chi_V \rangle = 2^n \cdot \mathbb{E}_{x \in \{\pm 1\}^n}[\chi_U(x)\chi_V(x)] = 2^n \prod_{i \in U \triangle V} \mathbb{E}[x_i]$$

which is equal to 1 if and only if $U = V$. Thus $B$ is an orthonormal basis for the set of Boolean functions and so we can conclude the following:

**Theorem A.1** (Fourier Expansion of Boolean Functions). *Let $f : \{\pm 1\}^n \to \mathbb{R}$ be boolean function. Then we can write $f$ uniquely as a multilinear polynomial Fourier Expansion:*

$$f = \sum_{U \subseteq [n]} \hat{f}_U \cdot \chi_U$$

*where $\hat{f}_U := \langle f, \chi_U \rangle$ are the Fourier Coefficients of $f$. The degree $\deg(f)$ of $f$ is defined as the largest cardinality $U$ for which $\hat{f}_U \neq 0$:*

$$\deg(f) := \max\{|U| : \hat{f}_U \neq 0\}$$

## A.1   INFLUENCE AND SENSITIVITY

Given $f : \{\pm 1\}^n \to \mathbb{R}$, we define can define its sensitivity by considering fluctuations in its output when one single bit is pertrubed:

**Definition A.2** (Influence). We define the **influence** of a coordinate $i$ as:

$$\text{Inf}_i[f] := \mathbb{E}_x \left[ \left( \frac{f(x) - f(x^{\oplus i})}{2} \right)^2 \right]$$

where $x^{\oplus i}$ is $x$ with the $i$-th coordinate flipped. Then we define the **total influence** of $f$ as:

$$I[f] := \sum_{i=1}^n \text{Inf}_i[f]$$

We can often think of total influence as a measure of **average sensitivity**.

The following lemma gives us a Fourier representation of influence:

**Lemma 2.** *If $f = \sum \hat{f}_S \chi_S$ then:*

$$\text{Inf}_i[f] = \sum_{S \ni i} \hat{f}(S)^2 \quad \text{and} \quad I[f] = \sum_{S \subseteq [n]} |S| \cdot \hat{f}(S)^2 = \sum_{k=0}^n k \cdot W^k[f]$$

*where $W^k[f] = \sum_{|S|=k} \hat{f}(S)^2$.*

The influence is related to other important quantities about $f$: its variance and its degree.

**Lemma 3** (Poincaré's inequality). *The **variance** of $f$ is:*

$$\text{Var}[f] = \mathbb{E}[f^2] - \mathbb{E}[f]^2 = \sum_{S \neq \emptyset} \hat{f}(S)^2$$

*It is true that:*

$$\text{Var}[f] \leq I[f]$$

The relationship between total influence and degree is very important because it allows us to represent functions with low total influence as low degree polynomials.

## B   MORE EXPERIMENTAL RESULTS WITH TOTAL INFLUENCE

In this section we provide additional experimental details from examining the total influence of four widely used models: GPT2, GEMMA-2-2B, RoBERTA, and BERT. As before, we run our experiments on $n = 256$ tokens and experiment both with $\mu$ being the uniform distribution and the distribution the model learns.

We analyze the geometric influence of these models by using forward hooks in Pytorch and collecting the gradients with respect to the input embeddings. We analyze positional influence both across layers and different attention heads, finding that in deep Transformers there are often layers with minimal influence towards the output. Such sparsity is an interesting phenomenon that could warrant further investigation.

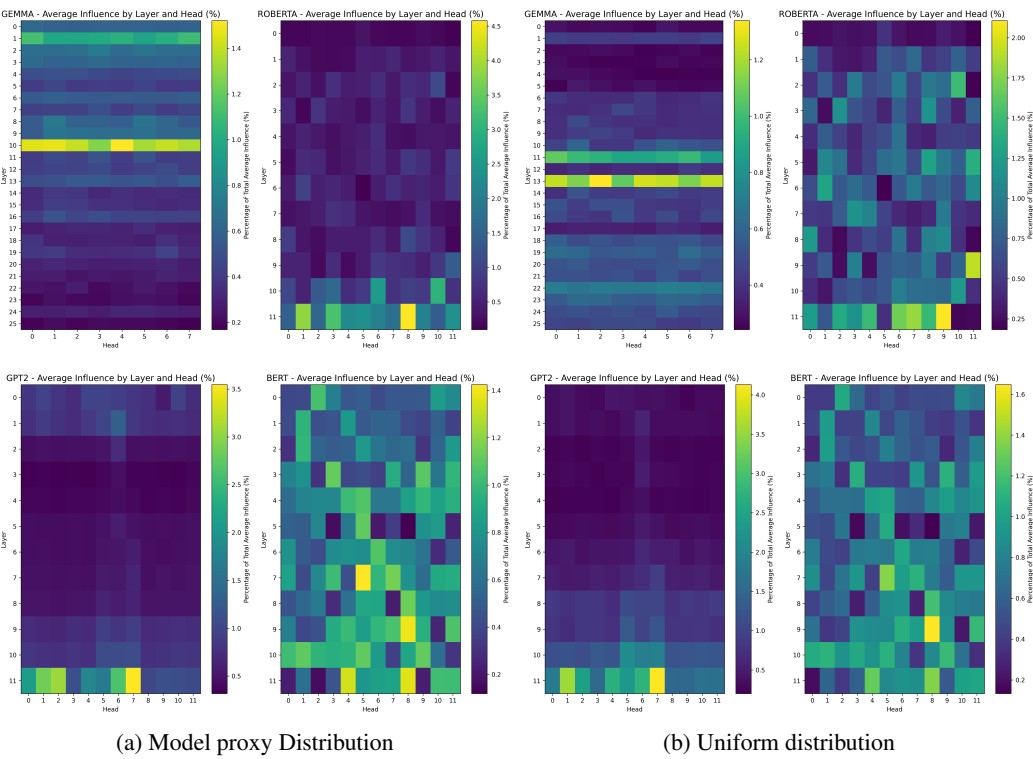

(a) Model proxy Distribution                    (b) Uniform distribution

Figure 7: Per Layer Geometric Influence Heatmaps for 4 different models: We observe that GEMMA-2-B has a very influential middle layer, while for ROBERTA and GPT2 the few layers are more influential.

## C   BASICS OF ORNSTEIN-UHLENBECK THEORY

We provide a self-contained review of the Ornstein-Uhlenbeck (OU) theory, culminating in the derivation of noise stability under the Gaussian measure. We encourage the interested reader to consult the excellent monograph of Andersson & Sjögren (2012) for more details.

### C.1   HERMITE POLYNOMIAL BASIS AND ITS PROPERTIES

The Ornstein-Uhlenbeck theory starts by considering the Gaussian measure $d\gamma(x) = \frac{1}{\pi^{d/2}} e^{-|x|^2} dx$ in $\mathbb{R}^d$. Let $L^2(\gamma)$ be the space of square-integrable functions with respect to this measure. Then, the *Physicist's Hermite Polynomials* turn out to be an invaluable way to spectrally decompose functions in this space.

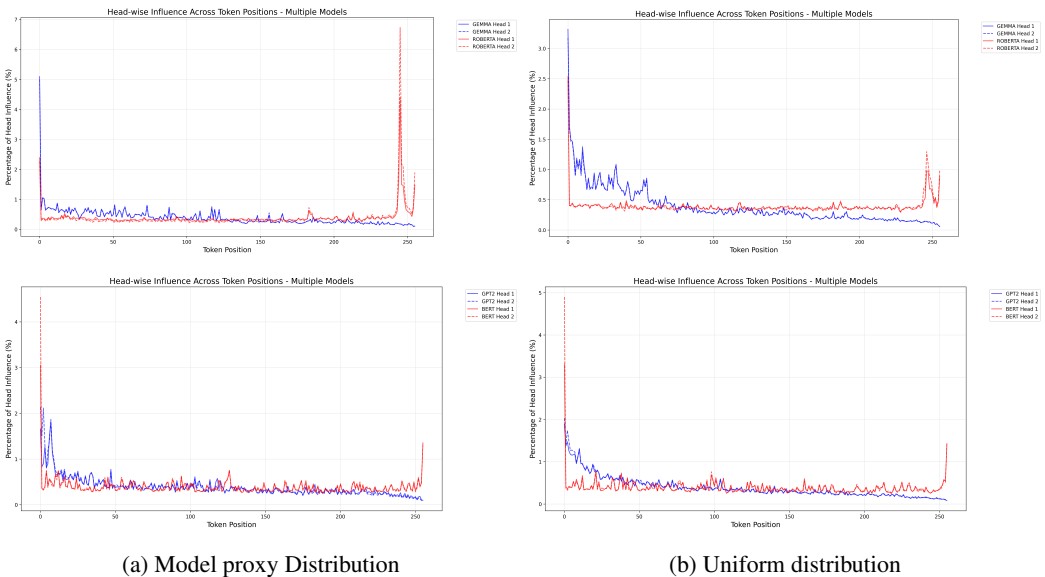

(a) Model proxy Distribution  (b) Uniform distribution

Figure 8: Geometric Influence Plots for two attention heads in 4 different models. We again observe a junta-like spectral concentration phenomenon in every model. The first and last token positions are consistently the most influential.

**Definition C.1** (Physicist's Hermite Polynomials). We define $H_0 = 1$ and:

$$H_n(x) = (-1)^n e^{x^2} \frac{d^n}{dx^n} e^{-x^2}$$

Some important properties of $H_n$ are captured by the Lemma below:

**Lemma 4** (Properties of $H_n$). *The following identities hold:*

$$\frac{d}{dx} H_n(x) = 2n H_{n-1}(x)$$
$$H_{n+1}(x) = 2x H_n(x) - 2n H_{n-1}(x)$$

*Also $H_n(x)$ is a degree $n$ polynomial with leading coefficient $2^n$.*

*Proof.* Let $D = \frac{d}{dx}$. We have by the product rule that:

$$DH_n(x) = (-1)^n e^{x^2} 2x D^n e^{-x^2} + (-1)^n e^{x^2} D(D^n e^{-x^2})$$

Then the generalized Leibniz formula gives:

$$DH_n(x) = (-1)^n e^{x^2} 2x D^n e^{-x^2} + (-1)^n e^{x^2} \sum_{k=0}^{n} \binom{n}{k} D^k (-2x) \cdot D^{n-k} e^{-x^2}$$

Only the terms where $k = 0, 1$ are non-zero, so we have:

$$DH_n(x) = (-1)^n e^{x^2} 2x D^n e^{-x^2} + (-1)^n e^{x^2} (-2x D^n e^{-x^2} + (-2)n D^{n-1} e^{-x^2} = 2n H_{n-1}(x)$$

Now, again by expanding $DH_n(x)$ we get:

$$DH_n(x) = (-1)^n e^{x^2} \frac{d^{n+1}}{dx^{n+1}} e^{-x^2} + (-1)^n e^{x^2} 2x \frac{d^n}{dx^n} e^{-x^2}$$
$$= -H_{n+1}(x) + 2x H_n(x)$$

And this gives:

$$H_{n+1}(x) = 2x H_n(x) - 2n H_{n-1}(x)$$

This allows us, by simple induction, to show that the leading coefficient of $H_n(x)$ is equal to $2^n$.  □

The most important statement in the Ornstein-Uhlenbeck theory is the following theorem:

**Theorem C.2** (Hermite Orthogonal Basis). *The set $B = \{H_n\}_{n=0}^{\infty}$ forms a complete orthogonal basis of $L^2(\gamma)$ with $||H_n||_{L^2(\gamma)} = 2^{n/2}\sqrt{n!}$. This motivates the definition and occasional use of the normalized Hermite basis:*

$$h_n(x) = \frac{1}{\sqrt{2^n n!}} H_n(x)$$

*which is orthonormal in $L^2(\gamma)$.*

*Proof.* To establish orthogonality, we have by repeated integration by parts that:

$$\int_{-\infty}^{\infty} H_n(x)H_m(x)d\gamma(x) = \frac{(-1)^n}{\sqrt{\pi}} H_m(x)e^{x^2}(D^n e^{-x^2})e^{-x^2}dx$$

$$= \frac{(-1)^n}{\sqrt{\pi}} \int_{-\infty}^{\infty} H_m(x) \cdot D^n(e^{-x^2})dx$$

$$= \frac{(-1)^n}{\sqrt{\pi}} \int_{-\infty}^{\infty} D^n H_m(x)e^{-x^2}dx$$

If $m < n$ then $D^n H_m(x) = 0$ and that gives orthogonality. For $m = n$, because the leading coefficient is $2^n$ we have:

$$||H_n||_{L^2(\gamma)} = 2^{n/2}\sqrt{n!}$$

as desired.

Finally, we have to establish completeness. Recall from analysis that completeness means for every $f \in L^2(\gamma)$ and every $\varepsilon > 0$ there must exist some $g \in \text{span}(B)$ such that $||f - g||_{L^2(\gamma)} \leq \varepsilon$. In other words, we seek to show that the linear span of $B$ is dense in $L^2(\gamma)$, or, in even different words, that the closure of the linear span is the whole space $L^2(\gamma)$. It suffices to show that if $f \in L^2(\gamma)$ is such that $\langle f, H_n \rangle = 0$ for all $n \in \mathbb{N}$ then $f = 0$. To see this, let us consider the spectral expansion of $f$ with the following coefficients:

$$\hat{f}(n) := \int f(x)H_n(x)d\gamma(x) = 0$$

By Parseval's identity, we have that:

$$||f||^2 = \sum_{n=0}^{\infty} \frac{|\hat{f}(n)|^2}{2^n n!} = 0$$

And thus $f = 0$, which concludes the proof. $\qquad\square$

When considering $d > 1$, we define the Hermite basis as a tensor product over multi-indices:

$$\boxed{H_a(x) = \prod_{i=1}^{d} H_{a_i}(x), a \in \mathbb{N}^d}$$

The same theorem about orthogonality and completeness holds in high dimensions as well.

## C.2   THE ORNSTEIN-UHLENBECK OPERATOR AND SEMIGROUP

Next, we will define the Ornstein-Uhlenbeck operator. This operator is an analog to the Laplacian in the $L^2(\gamma)$ space. It uses the *adjoint*[5] of the partial derivative operator $\partial_i := \frac{\partial}{\partial x_i}$:

$$\langle \partial_i f, g \rangle = \frac{1}{\pi^{d/2}} \int_{-\infty}^{\infty} \partial_i f(x)g(x)e^{-|x|^2}dx$$

$$= \frac{1}{\pi^{d/2}} \int_{-\infty}^{\infty} f(x)[2x_i g(x) - \partial_i g(x)]e^{-|x|^2}dx \qquad \text{(Integration by parts)}$$

$$= \langle f, (2x_i - \partial_i)g \rangle_{L^2(\gamma)}$$

Thus we arrive at the following definition:

---

[5]Recall the adjoint of a linear operator is an operator that moves to the other side of the inner product: $\langle Tx, y \rangle = \langle x, T^*y \rangle$.

**Definition C.3** (Ornstein-Uhlenbeck Operator). The **OU Operator** is defined as:

$$L = \frac{1}{2} \sum_{i=1}^{d} \partial_i^* \partial_i = -\frac{1}{2} \nabla + x\Delta$$

This operator is symmetric and positive and it has the wonderful property that the Hermite basis are actually its eigenfunctions:

**Theorem C.4** (Eigenfunctions of the OU operator). *The set $\{H_n\}_{n=0}^{\infty}$ are eigenfunctions of $L$:*

$$LH_\alpha = |a|H_\alpha$$

*where $|a| = \sum_{i=1}^{d} \alpha_i$*

*Proof.* Consider $d = 1$. We have for $j \neq n$:

$$\langle D^* H_{n-1}, H_j \rangle = 2j \langle H_{n-1}, H_{j-1} \rangle = 0$$

And if $j = n$ we have $\langle D^* H_{n-1}, H_n \rangle = 2nn!$, so $D^* H_{n-1} = H_n$. Thus we have showed that:

$$D^* H_{\alpha - e_i} = H_\alpha$$

for all $d \geq 1$ and $i \in [d]$. So we know how the adjoint partial operator acts on the Hermite polynomials. Thus, we can figure out how the OU operator acts on $H_\alpha$:

$$LH_\alpha = \frac{1}{2} \sum_{i=1}^{d} \partial_i^* \partial_i H_\alpha = \frac{1}{2} \sum_{i=1}^{d} \partial_i^* 2\alpha_i H_{\alpha - e_i} = \sum_{i=1}^{d} \alpha_i H_\alpha = |a|H_\alpha$$

as claimed. $\square$

Now we can define the OU Semigroup. A semigroup is a sequence of operators $T_t$ that describe the evolution of a process such that $T_0 = I$ and $T(t + s) = T(t) \cdot T(s)$[6]

**Definition C.5** (The OU Semigroup). The **OU Semigroup** is defined as:

$$(T_t)_{t \geq 0} = e^{-tL}$$

where if $f = \sum_\alpha \hat{f}(\alpha) H_\alpha$, $T_t$ acts on $f$ as:

$$e^{-tL} f = \sum_{\alpha \in \mathbb{N}^d} e^{-t|\alpha|} \hat{f}(\alpha) H_\alpha$$

### C.3 THE MEHLER KERNEL

It is easy to verify that $(T_t)$ is indeed a semigroup. A very useful tool for us to analyze properties of certain semigroups is **kernels**. If a semigroup is written via a kernel it will be very easy to prove powerful properties on it. A kernel is just the analogs of matrix multiplication. When operator $T$ acts on function $f$, imagine there being some kind of function $K(x, y)$ that for each $f(y)$ tells us how much that value "contributes" to $Tf(x)$:

$$Tf(x) = \int_{\mathbb{R}^d} K(x, y) f(y) dy$$

We want to find a kernel for the OU semigroup:

$$T_t f(x) = \int_{\mathbb{R}^d} M_t^\gamma(x, y) f(y) d\gamma(y)$$

Mehler already found this kernel (Fuchs & Hensel, 1866) in 1866!

---

[6]Technically there are also some analytic continuity requirements, but we will not dive into them here. Please refer to (Andersson & Sjögren, 2012) for more details.

**Theorem C.6** (The Mehler Kernel). *The Kernel for the OU semigroup is:*

$$M_t^\gamma(x,y) = \sum_{\alpha \in \mathbb{N}^d} e^{-t|\alpha|} h_\alpha(x) h_\alpha(y)$$

*Proof.* We shall just verify this. Choose some $\beta$ and see how the kernel acts on $H_\beta$:

$$\int_{y \in \mathbb{R}^d} \sum_{|\alpha|<N} e^{-t|\alpha|} h_\alpha(x) h_\alpha(y) H_\beta(y) d\gamma(y) = \sum_{|\alpha|<N} e^{-t|\alpha|} h_\alpha(x) \int_{y \in \mathbb{R}^d} h_\alpha(y) H_\beta(y) d\gamma(y)$$

$$= e^{-t|\beta|} \langle h_\beta, H_\beta \rangle h_\alpha(y) \qquad \text{(only } \alpha = \beta \text{ survives)}$$

$$= e^{-t|\beta|} ||H_\beta|| h_\beta(x)$$

$$= T_t H_\beta$$

Now take $N \to \infty$ and we arrive at the correct result. $\qquad \square$

The Mehler kernel has a really nice analytical expression. Starting from $H_n(y) = (-1)^n e^{-y^2} D^n e^{-y^2}$ and considering the Fourier Transform of the Gaussian: $\mathcal{F}(e^{-\xi^2})(x) = \sqrt{pi} \cdot e^{-x^2/4}$ we have:

$$H_n(y) = (-1)^n e^{y^2} \frac{2^n i^n}{\sqrt{\pi}} \int_{-\infty}^\infty \xi^n e^{2iy\xi - \xi^2} d\xi$$

And so the Mehler kernel can be written[7] as:

$$M_t^\gamma(x,y) = \sum_{n=0}^\infty e^{-tn} h_n(x) h_n(y)$$

$$= \sum_{n=0}^\infty e^{-tn} h_n(x) (-1)^n e^{y^2} \frac{2^n i^n}{\sqrt{\pi}} \int_{-\infty}^\infty \xi^n e^{2iy\xi - \xi^2} d\xi$$

$$= \frac{e^{y^2}}{\sqrt{\pi}} \int_{-\infty}^\infty \sum_{n=0}^\infty \frac{1}{n!} (-i\xi e^{-t})^n H_n(x) e^{2iy\xi - \xi^2} d\xi \qquad \text{(Switch } \sum \text{ and } \int)$$

$$= \frac{e^{y^2}}{\sqrt{\pi}} \int_{-\infty}^\infty e^{2i\xi(y - e^{-t}x) - \xi^2(1 - e^{-2t})} d\xi$$

Letting $\xi' = \xi\sqrt{1 - e^{-2t}}$ and taking the inverse Fourier Transform we get that:

$$M_t^\gamma(x,y) = \frac{1}{\sqrt{\pi}} \frac{e^{y^2}}{\sqrt{1 - e^{-2t}}} \int_{-\infty}^\infty e^{2i\xi' \frac{y - e^{-t}x}{\sqrt{1 - e^{-2t}}} - |\xi'|^2} d\xi$$

And this brings us to the following important theorem:

**Theorem C.7** (Analytical form of Mehler's Kernel). *We have that:*

$$M_t^\gamma(x,y) = \frac{1}{\pi^{d/2}(1 - e^{-2t})^{d/2}} e^{-\frac{|y - e^t x|^2}{1 - e^{-2t}}}$$

Some important consequences of this treatment is the easy, via Hölder's inequality, proof of the non-expansiveness of $T_t$:

**Lemma 5** (Non-expansiveness of OU operator). *For any $p \geq 1$ we have that:*

$$||T_t||^p_{L^p(\gamma)} \leq ||T_t|f|^p||_{L^p(\gamma)} \leq ||f||^p_{L^p(\gamma)}$$

We also easily get *Mehler's formula*, which will be the starting point in our stability argument. Just perform the change of variables

$$z = \frac{y - e^{-t}x}{\sqrt{1 - e^{-2t}}}$$

---

[7] Formally, we also need to justify why switching infinite summation and integration is possible by using dominated convergence, but we will skip this here.

to get:

$$T_t f(X) = \int f\left(\rho X + Z\sqrt{1-\rho^2}\right) d\gamma(Z)$$

where $\rho := e^{-t}$ for $t \geq 0$. In other words,

$$T_t f(X) = \mathbb{E}_{Z \sim \mathcal{N}(0,1)}[f(\rho X + Z\sqrt{1-\rho^2})]$$

which allows us to view $T_t$ as the expected outcome of a random process: take $X$, add some noise to it to get $Y = \rho X + Z\sqrt{1-\rho^2}$ and see the expected value of $f(Y)$.

## C.4 FROM OU THEORY TO STABILITY

We are finally ready to define stability in familiar terms:

**Definition C.8** (Gaussian Noise Stability). Let $f \in L^2(\gamma)$ and $X \sim \gamma^n$ where $\gamma \equiv \mathcal{N}(0,1)$. Let $\rho \in (0,1)$ and $Y = \rho X + Z\sqrt{1-\rho^2}$ for $Z \sim \mathcal{N}(0, I_n)$ independent from $X$. We define the stability of $f$ as:

$$\mathrm{Stab}_\rho(f) := \mathbb{E}_{(X,Y)}[f(X)f(Y)]$$

We note the immediate connection between stability and the OU semigroup:

$$\mathrm{Stab}_\rho(f) = \langle T_\rho f, f \rangle_{L^2(\gamma)}$$

Now we know from the spectral expansion of the OU Semigroup action that:

$$T_\rho f = \sum_{\alpha \in \mathbb{N}^d} \rho^{|\alpha|} \hat{f}(\alpha) H_\alpha$$

By orthogonality of the Hermite polynomials we have:

$$\mathrm{Stab}_\rho(f) = \left\langle \sum_{\alpha \in \mathbb{N}^d} \rho^{|\alpha|} \hat{f}(\alpha) H_\alpha, \sum_{\alpha \in \mathbb{N}^d} \rho^{|\alpha|} \hat{f}(\alpha) H_\alpha \right\rangle = \sum_{\alpha \in \mathbb{N}^d} \rho^{|\alpha|} 2^{|\alpha|} \cdot \alpha! \cdot \hat{f}(\alpha)^2$$

If we define $\tilde{f}(a) = \sqrt{2^{|\alpha|}\alpha!}\hat{f}(\alpha)$ we finally get:

$$\mathrm{Stab}_\rho(f) = \sum_{\alpha \in \mathbb{N}^d} \rho^{|\alpha|} \tilde{f}(\alpha)^2$$

## D PROOF OF LEMMA 1

**Lemma 6** (Spectral tail bound via stability). *Let $f \in L^2(\gamma^n)$ be a square-integrable function under the standard Gaussian measure $\gamma^n$. Suppose that*

$$Stab_\rho(f) := \mathbb{E}[f(X)f(Y)] \geq (1-\delta)\|f\|_2^2$$

*for some $\rho \in (0,1)$ and $0 < \delta < \varepsilon < 1$, where $(X,Y)$ are standard Gaussian vectors with correlation $\rho$, i.e., $Y = \rho X + \sqrt{1-\rho^2}Z$ for $Z \sim \gamma^n$ independent of $X$.*

*Then $f$ is $(\varepsilon, T)$-concentrated for*

$$\boxed{T \geq \frac{\log(1 - \delta/\varepsilon)}{\log \rho}}$$

*in the sense that:*

$$\sum_{k \geq T} \sum_{|\alpha|=k} \hat{f}(\alpha)^2 \cdot 2^{|\alpha|}\alpha! \leq \varepsilon\|f\|_2^2$$

*where $f(x) = \sum_{\alpha \in \mathbb{N}^n} \hat{f}(\alpha) H_\alpha(x)$ is the expansion of $f$ in the multivariate physicists' Hermite polynomial basis.*

*Proof.* Let us write the Hermite expansion of $f$ as

$$f(x) = \sum_{\alpha \in \mathbb{N}^n} \hat{f}(\alpha) H_\alpha(x)$$

where $H_\alpha(x) = \prod_{i=1}^{n} H_{\alpha_i}(x_i)$, and $H_k$ is the physicists' Hermite polynomial of degree $k$. The Hermite polynomials are orthogonal in $L^2(\gamma^n)$ with squared norm $\|H_\alpha\|^2 = 2^{|\alpha|}\alpha!$.

Parseval's identity in this basis reads:

$$\|f\|_2^2 = \sum_\alpha \hat{f}(\alpha)^2 \cdot 2^{|\alpha|}\alpha!$$

We define the Hermite level-$k$ weight of $f$ as:

$$W^{(k)}[f](f) := \sum_{|\alpha|=k} \hat{f}(\alpha)^2 \cdot 2^{|\alpha|}\alpha!$$

With this notation, Parseval's identity is simply $\|f\|_2^2 = \sum_{k=0}^{\infty} W^{(k)}[f](f)$.

The Ornstein–Uhlenbeck semigroup $T_\rho$ acts on the Hermite expansion by

$$T_\rho f(x) = \sum_\alpha \rho^{|\alpha|} \hat{f}(\alpha) H_\alpha(x)$$

The noise stability can therefore be written as:

$$\begin{aligned}
\text{Stab}_\rho(f) &= \langle f, T_\rho f \rangle \\
&= \sum_\alpha \rho^{|\alpha|} \hat{f}(\alpha)^2 \cdot 2^{|\alpha|}\alpha! \\
&= \sum_{k=0}^{\infty} \rho^k W^{(k)}[f](f)
\end{aligned}$$

Fix a threshold $T \in \mathbb{N}$. We can split the sum at degree $T$:

$$\text{Stab}_\rho(f) = \sum_{k<T} \rho^k W^{(k)}[f](f) + \sum_{k \geq T} \rho^k W^{(k)}[f](f)$$

Since $\rho \in (0,1)$, we can upper-bound the terms in the sums by $\rho^k \leq 1$ for $k < T$ and $\rho^k \leq \rho^T$ for $k \geq T$. This gives:

$$\begin{aligned}
\text{Stab}_\rho(f) &\leq \sum_{k<T} W^{(k)}[f](f) + \rho^T \sum_{k \geq T} W^{(k)}[f](f) \\
&= \left( \|f\|_2^2 - \sum_{k \geq T} W^{(k)}[f](f) \right) + \rho^T \sum_{k \geq T} W^{(k)}[f](f) \\
&= \|f\|_2^2 - (1 - \rho^T) \sum_{k \geq T} W^{(k)}[f](f)
\end{aligned}$$

Rearranging the inequality to isolate the tail sum, we have:

$$\sum_{k \geq T} W^{(k)}[f](f) \leq \frac{\|f\|_2^2 - \text{Stab}_\rho(f)}{1 - \rho^T}$$

Using the assumption that $\text{Stab}_\rho(f) \geq (1 - \delta)\|f\|_2^2$, we can further bound the numerator:

$$\sum_{k \geq T} W^{(k)}[f](f) \leq \frac{\|f\|_2^2 - (1 - \delta)\|f\|_2^2}{1 - \rho^T} = \frac{\delta \|f\|_2^2}{1 - \rho^T}$$

To ensure this tail sum is at most $\varepsilon \|f\|_2^2$, it suffices to have:

$$\frac{\delta \|f\|_2^2}{1 - \rho^T} \leq \varepsilon \|f\|_2^2$$

Assuming $f$ is not the zero function, we can cancel $\|f\|_2^2$ and solve for $T$:

$$T \log \rho \leq \log \left( 1 - \frac{\delta}{\varepsilon} \right)$$

$$T \geq \frac{\log \left( 1 - \delta/\varepsilon \right)}{\log \rho}$$

This bound is well-defined provided the argument of the logarithm is positive, which holds due to the assumption that $\delta < \varepsilon$. $\qquad\square$

## E   PROOF OF THEOREM 5.1

**Theorem E.1** (Stability of ReLU). *Let* $(X, Y) \sim \mathcal{N} \left( 0, \begin{bmatrix} 1 & \rho \\ \rho & 1 \end{bmatrix} \right)$ *for* $\rho \in (-1, 1)$*. Then:*

$$\mathbb{E}[ReLU(X)ReLU(Y)] = \frac{1}{2\pi} \left( \sqrt{1 - \rho^2} + \rho(\pi - \arccos \rho) \right)$$

*Proof.* We compute the expectation:

$$I(\rho) := \mathbb{E}[\text{ReLU}(X)\text{ReLU}(Y)] = \mathbb{E}[\max(0, X) \max(0, Y)].$$

The joint density function of $(X, Y)$ is

$$f(x, y) = \frac{1}{2\pi\sqrt{1 - \rho^2}} \exp\left( -\frac{x^2 - 2\rho xy + y^2}{2(1 - \rho^2)} \right).$$

Thus, we can compute:

$$I(\rho) = \int_0^\infty \int_0^\infty xy \cdot f(x, y) \, dy \, dx.$$

We perform a change to polar coordinates $x = r\cos\theta$, $y = r\sin\theta$, $r \in [0, \infty), \theta \in [0, \pi/2]$ so that $x, y \geq 0$. Then the Jacobian is $dx \, dy = r \, dr \, d\theta$ and the exponent becomes:

$$\frac{x^2 - 2\rho xy + y^2}{2(1 - \rho^2)} = \frac{r^2(\cos^2\theta - 2\rho\cos\theta\sin\theta + \sin^2\theta)}{2(1 - \rho^2)} \qquad \text{(Substitute polar coordinates)}$$

$$= \frac{r^2(1 - \rho\sin(2\theta))}{2(1 - \rho^2)} \qquad (\sin(2\theta) = 2\sin\theta\cos\theta)$$

So the integral becomes:

$$I(\rho) = \int_0^{\pi/2} \int_0^\infty r^2 \cos\theta \sin\theta \cdot \frac{1}{2\pi\sqrt{1 - \rho^2}} \exp\left( -\frac{r^2(1 - \rho\sin(2\theta))}{2(1 - \rho^2)} \right) r \, dr \, d\theta$$

$$= \frac{1}{2\pi\sqrt{1 - \rho^2}} \int_0^{\pi/2} \cos\theta \sin\theta \int_0^\infty r^3 \exp\left( -\frac{r^2(1 - \rho\sin(2\theta))}{2(1 - \rho^2)} \right) dr \, d\theta.$$

Now let

$$a(\theta) = \frac{1 - \rho\sin(2\theta)}{2(1 - \rho^2)}.$$

Use the substitution $u = r^2 \Rightarrow du = 2r \, dr$, so:

$$\int_0^\infty r^3 e^{-a(\theta)r^2} \, dr = \frac{1}{2} \int_0^\infty u e^{-a(\theta)u} \, du = \frac{1}{2} \cdot \frac{1}{a(\theta)^2} \qquad \text{(Standard integral: } \int_0^\infty u e^{-au} du = \frac{1}{a^2}\text{)}$$

So we obtain:

$$I(\rho) = \frac{1}{2\pi\sqrt{1-\rho^2}} \int_0^{\pi/2} \cos\theta \sin\theta \cdot \frac{1}{2a(\theta)^2} \, d\theta$$

$$= \frac{1}{4\pi\sqrt{1-\rho^2}} \int_0^{\pi/2} \frac{\cos\theta \sin\theta}{a(\theta)^2} \, d\theta.$$

Now recall that:

$$a(\theta) = \frac{1 - \rho\sin(2\theta)}{2(1-\rho^2)} \Rightarrow a(\theta)^2 = \frac{(1 - \rho\sin(2\theta))^2}{4(1-\rho^2)^2}$$

So:

$$\frac{1}{a(\theta)^2} = \frac{4(1-\rho^2)^2}{(1 - \rho\sin(2\theta))^2}$$

Plugging in:

$$I(\rho) = \frac{1}{4\pi\sqrt{1-\rho^2}} \int_0^{\pi/2} \cos\theta \sin\theta \cdot \frac{4(1-\rho^2)^2}{(1 - \rho\sin(2\theta))^2} \, d\theta$$

$$= \frac{(1-\rho^2)^{3/2}}{\pi} \int_0^{\pi/2} \frac{\cos\theta \sin\theta}{(1 - \rho\sin(2\theta))^2} \, d\theta.$$

Now make the substitution $u = \tan\theta$, so:

$$\sin(2\theta) = \frac{2u}{1+u^2}, \quad \cos\theta \sin\theta \, d\theta = \frac{u}{(1+u^2)^2} \, du$$

Thus:

$$\int_0^{\pi/2} \frac{\cos\theta \sin\theta}{(1 - \rho\sin(2\theta))^2} \, d\theta = \int_0^\infty \frac{u}{(1+u^2)^2 \left(1 - \rho \cdot \frac{2u}{1+u^2}\right)^2} \, du = \int_0^\infty \frac{u}{(u^2 + 1 - 2\rho u)^2} \, du.$$

This integral can be evaluated by completing the square:

$$u^2 - 2\rho u + 1 = (u - \rho)^2 + (1 - \rho^2)$$

So, now consider the integral

$$I := \int_0^\infty \frac{u}{\left((u-\rho)^2 + (1-\rho^2)\right)^2} \, du,$$

where we define

$$a := \sqrt{1 - \rho^2}.$$

We rewrite the numerator as

$$I = \int_0^\infty \frac{(u-\rho) + \rho}{\left((u-\rho)^2 + a^2\right)^2} \, du$$

$$= \int_0^\infty \frac{u - \rho}{\left((u-\rho)^2 + a^2\right)^2} \, du + \rho \int_0^\infty \frac{1}{\left((u-\rho)^2 + a^2\right)^2} \, du$$

$$= I_1 + \rho I_2.$$

**Integral $I_1$:** Substitute $v = u - \rho$, then $du = dv$, and the integration limits become $v = -\rho$ to $\infty$:

$$I_1 = \int_{-\rho}^{\infty} \frac{v}{(v^2 + a^2)^2} \, dv.$$

Using the antiderivative

$$\frac{d}{dv}\left(\frac{1}{v^2 + a^2}\right) = -\frac{2v}{(v^2 + a^2)^2},$$

we have

$$\int \frac{v}{(v^2 + a^2)^2} \, dv = -\frac{1}{2(v^2 + a^2)} + C.$$

Evaluating at the limits:

$$I_1 = \left[-\frac{1}{2(v^2 + a^2)}\right]_{v=-\rho}^{v\to\infty} = \lim_{M\to\infty}\left(-\frac{1}{2(M^2 + a^2)} + \frac{1}{2(\rho^2 + a^2)}\right)$$

$$= 0 + \frac{1}{2(\rho^2 + a^2)} = \frac{1}{2 \cdot 1} = \frac{1}{2},$$

since $\rho^2 + a^2 = \rho^2 + (1 - \rho^2) = 1$.

**Integral $I_2$:** Again substitute $v = u - \rho$, so

$$I_2 = \int_{-\rho}^{\infty} \frac{1}{(v^2 + a^2)^2} \, dv.$$

The antiderivative is known:

$$\int \frac{dv}{(v^2 + a^2)^2} = \frac{v}{2a^2(v^2 + a^2)} + \frac{1}{2a^3}\arctan\left(\frac{v}{a}\right) + C.$$

Evaluating at the limits:

$$I_2 = \lim_{M\to\infty}\left(\frac{M}{2a^2(M^2 + a^2)} + \frac{1}{2a^3}\arctan\left(\frac{M}{a}\right)\right)$$
$$- \left(\frac{-\rho}{2a^2(\rho^2 + a^2)} + \frac{1}{2a^3}\arctan\left(\frac{-\rho}{a}\right)\right).$$

Since

$$\lim_{M\to\infty}\frac{M}{2a^2(M^2 + a^2)} = 0, \quad \text{and} \quad \lim_{M\to\infty}\arctan\left(\frac{M}{a}\right) = \frac{\pi}{2},$$

we get

$$I_2 = \frac{\pi}{4a^3} + \frac{\rho}{2a^2} - \frac{1}{2a^3}\arctan\left(-\frac{\rho}{a}\right).$$

Using the oddness of arctangent,

$$\arctan(-x) = -\arctan(x),$$

we rewrite

$$I_2 = \frac{\pi}{4a^3} + \frac{\rho}{2a^2} + \frac{1}{2a^3}\arctan\left(\frac{\rho}{a}\right).$$

**Combining $I_1$ and $I_2$:**

$$I = I_1 + \rho I_2 = \frac{1}{2} + \rho\left(\frac{\pi}{4a^3} + \frac{\rho}{2a^2} + \frac{1}{2a^3}\arctan\left(\frac{\rho}{a}\right)\right).$$

Finally, recall the trigonometric identity:

$$\arctan\left(\frac{\rho}{a}\right) = \frac{\pi}{2} - \arccos\rho,$$

which allows for the final form after multiplying back by the outer constant $(1 - \rho^2)^{3/2}/\pi$, we get:

$$\mathbb{E}[\text{ReLU}(X)\text{ReLU}(Y)] = \frac{1}{2\pi}\left(\sqrt{1 - \rho^2} + \rho(\pi - \arccos\rho)\right).$$

$$\square$$

### E.1 APPROXIMATING WITH A QUADRATIC

The expression we gave above for the stability propagation is hard to evaluate. We can approximate it with a quadratic really well:

**Lemma 7** (Quadratic Approximation of $f(\rho)$). *Define the function*

$$s(\rho) := \frac{1}{2\pi} \left( \sqrt{1 - \rho^2} + \rho\big(\pi - \arccos \rho\big) \right)$$

*for $\rho \in (-1, 1)$. Then the quadratic approximation of $s$ around $\rho = 0$ is*

$$s(\rho) = \frac{1}{2\pi} + \frac{1}{4}\rho + \frac{1}{4\pi}\rho^2 + o(\rho^2).$$

*Proof.* First, evaluate the function at zero:

$$s(0) = \frac{1}{2\pi} \left( \sqrt{1 - 0^2} + 0 \cdot (\pi - \arccos 0) \right) = \frac{1}{2\pi}.$$

Next, compute the first derivative:

$$s'(\rho) = \frac{1}{2\pi} \left( -\frac{\rho}{\sqrt{1 - \rho^2}} + \pi - \arccos \rho - \rho \cdot \frac{d}{d\rho} \arccos \rho \right).$$

Using $\frac{d}{d\rho} \arccos \rho = -\frac{1}{\sqrt{1-\rho^2}}$, this simplifies to

$$s'(\rho) = \frac{1}{2\pi} \left( \pi - \arccos \rho \right).$$

Evaluating at $\rho = 0$:

$$s'(0) = \frac{1}{2\pi} \left( \pi - \frac{\pi}{2} \right) = \frac{1}{4}.$$

Now compute the second derivative:

$$s''(\rho) = \frac{1}{2\pi} \frac{d}{d\rho} (\pi - \arccos \rho) = \frac{1}{2\pi} \frac{1}{\sqrt{1 - \rho^2}},$$

and at $\rho = 0$:

$$s''(0) = \frac{1}{2\pi}.$$

Therefore, the quadratic approximation at $\rho = 0$ is

$$s(\rho) \approx s(0) + S'(0)\rho + \frac{s''(0)}{2}\rho^2 = \frac{1}{2\pi} + \frac{1}{4}\rho + \frac{1}{4\pi}\rho^2.$$

$\square$

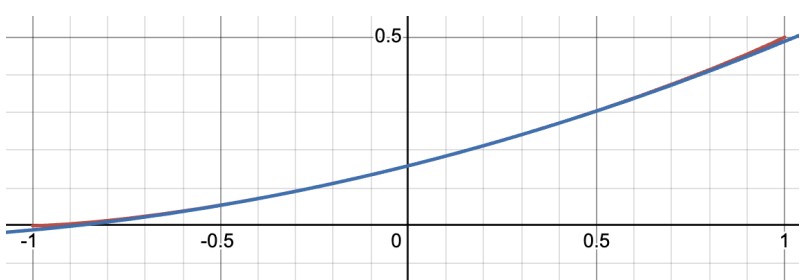

Figure 9: Approximating the stability of MLP with a quadratic

# F   PROOF OF THEOREM 5.2

## F.1   PRELIMINARIES

We will make use of the following mathematical tools in the proofs below:

**Lemma 8** (Laurant-Massart Concentration Bounds for $\chi_d^2$ Random Variables). *Let $Z$ be a chi-squared random variable with $d$ degrees of freedom. Then:*

$$\Pr\left[|Z - d| \geq 2\sqrt{du} + 2u\right] \leq e^{-u}$$

**Lemma 9** (Concentration of Gaussian Random Variable). *Let $Z \sim \mathcal{N}(0, \sigma^2)$. Then:*

$$\Pr\left[Z \geq \alpha\right] \leq e^{-\alpha^2/(2\sigma^2)}$$

**Proposition 1** (Weighted Cauchy-Schwarz Inequality). *Let $a_1, ..., a_n$ be non-negative constants and $y_1, ..., y_n \in \mathbb{R}^d$. It is true that:*

$$\left\|\sum_{j=1}^n a_j y_j\right\|_2 \leq \left(\sum_{j=1}^n a_j\right)^{1/2} \cdot \left(\sum_{j=1}^n a_j \|y_j\|_2^2\right)^{1/2}$$

*Proof.* First, by the triangle inequality for the $\ell_2$-norm and the fact that $a_j \geq 0$, we have:

$$\left\|\sum_{j=1}^n a_j y_j\right\|_2^2 \leq \left(\sum_{j=1}^n \|a_j y_j\|_2\right)^2 = \left(\sum_{j=1}^n a_j \|y_j\|_2\right)^2 = \left(\sum_{j=1}^n \sqrt{a_j} \cdot \sqrt{a_j} \cdot \|y_j\|_2\right)^2$$

Next, we apply the Cauchy-Schwarz inequality to the term on the right-hand side, to obtain:

$$\left(\sum_{j=1}^n \sqrt{a_j} \cdot (\sqrt{a_j} \|y_j\|_2)\right)^2 \leq \left(\sum_{j=1}^n a_j\right)\left(\sum_{j=1}^n a_j \|y_j\|_2^2\right)$$

which finalizes the proof.   $\square$

## F.2   PROVING THE THEOREM

**Theorem F.1** (Stability of Attention Layer, $W_Q W_K^T = I_d$). *Let $X \sim \mathcal{N}(0, I_{n \times d})$ and $Y = \rho X + Z\sqrt{1 - \rho^2}$ for $Z \sim \mathcal{N}(0, I_{n \times d}) \perp X$. Let $f : \mathbb{R}^{n \times d} \to \mathbb{R}^{n \times d}$ where $f(X) = \sigma(XX^T) \cdot XW_V$, $W_V \in \mathbb{R}^{d \times d}$ and $\sigma$ is the row-softmax function. Then for all $i \in [n]$ and $j \in [d]$ we have that:*

$$\lim_{d \to \infty} \mathbb{E}[f(X)_{ij} f(Y)_{ij}] = \rho \|(W_V)_{:,j}\|_2^2$$

*Proof.* Let $x_1, ..., x_n \in \mathbb{R}^d$ be the rows of $X$. We first show that $W := \sigma(XX^T)$ converges to $I_n$ in probability as $d \to \infty$ with exponential tails in $d$.

First, fix some $i \in [n]$ and let $\sigma^2 := \|x_i\|_2^2$. We know that $\sigma^2 \sim \chi_d^2$ and so by Lemma 8 we have that:

$$\Pr\left[\frac{23d}{32} \leq \sigma^2 \leq \frac{41d}{32}\right] \geq 1 - 2e^{-d/64}$$

Let $A$ be the likely event defined above. Let us condition on $A$ happening.

Now, condition on $x_i$ and let $V_j := \langle x_i, x_j \rangle - \sigma^2$. Since $x_i \perp x_j$ we have that $V_j \sim \mathcal{N}(-\sigma^2, \sigma^2) \mid x_i$ and so by Lemma 9 we have:

$$\Pr\left[V_j \geq -\frac{\sigma^2}{2} \mid x_i\right] = \Pr\left[\langle x_i, x_j \rangle \geq \frac{\sigma^2}{2} \mid x_i\right] \leq e^{-\frac{\sigma^4/4}{2\sigma^2}} = e^{-\sigma^2/8} \leq e^{-23d/256}$$

By a union bound, we have that:

$$\Pr\left[\exists j \neq i \text{ s.t. } V_j \geq -\frac{\sigma^2}{2} \mid x_i, A\right] \leq (n-1)e^{-\frac{23}{256}d}$$

Removing the conditioning on $x_i$ and $A$, we get that:

$$\Pr\left[\exists j \neq i \text{ s.t. } V_j \geq -\frac{\sigma^2}{2}\right] \leq \Pr[\neg A] + (n-1)e^{-\frac{23}{256}d} \leq 2e^{-d/64} + (n-1)e^{-\frac{23}{256}d}$$

Therefore, with probability at least $1 - e^{-\Theta(d)}$ it holds that $V_j \leq -\frac{\sigma^2}{2} \leq -\frac{23d}{64}$. Let us call this event $B$ and condition on it.

Now, we focus on the $i$-th row of $W$. For $j \neq i$, we have:

$$W_{ij} = \frac{\exp(V_j)}{1 + \sum_{k \neq i} \exp(V_k)} \leq \exp(V_j) \leq \exp\left(-\frac{23}{64}d\right)$$

Conditioning on $H_d$, we have proven that:

$$\Pr\left[\max_{j \neq i} W_{ij} > \exp\left(-\frac{41d}{64}\right)\right] \leq O\left(ne^{-\Theta(d)}\right)$$

In other words, $\boxed{W_{ij} \xrightarrow{p} 0 \text{ as } d \to \infty}$ with an exponential tail.

Similarly, when $i = j$, we have that:

$$W_{ii} = \frac{1}{1 + \sum_{k \neq i} \exp(V_k)} \leq \frac{1}{1 - (n-1)\exp(-\frac{23}{64}d)}$$

meaning that $\boxed{W_{ii} \xrightarrow{p} 1}$.

Now that we have shown that $W \xrightarrow{p} I_n$ as $d \to \infty$, let us consider the product $W_{i,:} \cdot XW_V$. We want to treat this product as $e_i XA$, so we show that the error goes to 0:

$$W_{i,:} \cdot XA - e_1 \cdot XW_V = \sum_{j \neq i} w_{ij} x_j W_V =: E_X$$

We have by the weighted Cauchy-Schwatz inequality that:

$$\mathbb{E}\left[||E_X||_2^2\right] \leq \left(\sum_{j \neq i} W_{ij}\right) \cdot \left(\sum_{j \neq i} W_{ij}\mathbb{E}||x_j W_V||_2^2\right)$$

$$\leq (n-1)\sum_{j \neq i} \mathbb{E}[M_d \cdot ||x_j W_V||_2^2]$$

where $M_d := \max_{j \neq i} W_{ij} \in [0, 1]$. We know that $M_d \to 0$ in probability, so by the dominated convergence theorem we get that $\mathbb{E}[M_d] \to 0$ as $d \to \infty$. Also, $\mathbb{E}\left[\sum_{j \neq i} ||x_j W_V||_2^2\right]$ is a constant with respect to $d$. Applying Cauchy-Schwartz again, we get:

$$\mathbb{E}[||E_X||_2^2] \leq (n-1)\sqrt{\mathbb{E}[M_d^2] \cdot \mathbb{E}\left[\sum_{j \neq i} ||x_j W_V||_2^2\right]} \xrightarrow[d \to \infty]{} 0$$

Similarly, we can also show that $E[||E_Y||_2^2] \to 0$ as $d \to \infty$.

With that, if we fix $j \in [d]$, let $\varepsilon_{X,i,j} := f(X)_{i,j} - (x_i A)_j$. We have:

$$\boxed{\mathbb{E}[\varepsilon_{X,i,j}^2] \leq \mathbb{E}[||E_X||_2^2] \to 0, \qquad \mathbb{E}[\varepsilon_{Y,i,j}^2] \leq \mathbb{E}[||E_Y||_2^2] \to 0}$$

We can therefore claim that:

$$\boxed{\mathbb{E}[f(X)_{i,j} \cdot f(Y)_{i,j}] = \mathbb{E}[(x_i W_V)_j \cdot (y_i W_V)_j] + o(1)}$$

To see this, observe that:

$$f(X)_{i,j} f(Y)_{i,j} - (x_i W_V)_j (y_i W_V)_j = (x_i W_V)_j \cdot \varepsilon_{X,i,j} + (y_i W_V)_j \cdot \varepsilon_{Y,i,j} + \varepsilon_{X,i,j} \cdot \varepsilon_{Y,i,j}$$

Each term has expectation $o(1)$:

- $\mathbb{E}[(x_i W_V)_j \cdot \varepsilon_{Y,i,j}] \leq \sqrt{\mathbb{E}[(x_i W_V)_j^2]} \cdot \sqrt{\mathbb{E}[\varepsilon_{Y,i,j}^2]} = ||(W_V)_{:,j}||_2 \cdot o(1) = o(1)$

- $\mathbb{E}[(y_i W_V)_j \cdot \varepsilon_{X,i,j}] = o(1)$ symmetrically.

- $\mathbb{E}[\varepsilon_{X,i,j} \cdot \varepsilon_{Y,i,j}] \leq \sqrt{\mathbb{E}[\varepsilon_{X,i,j}^2]} \cdot \sqrt{\mathbb{E}[\varepsilon_{Y,i,j}^2]} = o(1) \cdot o(1) = o(1).$

Finally, we can obtain the final calculation:

$$
\begin{aligned}
\lim_{d \to \infty} \mathbb{E}[f(X)_{i,j} f(Y)_{i,j}] &= \mathbb{E}[(x_i W_V)_j \cdot (y_i W_V)_j] + o(1) \\
&= \mathbb{E}[\langle x_i, (W_V)_{:,j} \rangle \cdot \langle y_i, (W_V)_{:,j} \rangle] + o(1) \\
&= \sum_{k,\ell} (W_V)_{k,j} (W_V)_{\ell,j} \cdot \mathbb{E}[x_{i,k} \cdot y_{i,\ell}] + o(1) \\
&= \rho \cdot \sum_{\ell} (W_V)_{\ell,j}^2 + o(1) \qquad \text{(By definition of } (X,Y)\text{)} \\
&= \rho \cdot ||(W_V)_{:,j}||_2^2 + o(1)
\end{aligned}
$$

as claimed. $\qquad \square$

## G  NOISE STABILITY PROPAGATION IN THE UNSTRUCTURED CASE

In this section we prove the following theorem:

**Theorem G.1.** *The noise stability of an attention layer in the unstructured case is:*

$$\lim_{d \to \infty} \mathbb{E}[f(X)_{ij} f(Y)_{ij}] \overset{p}{=} \rho \cdot s(\rho) \cdot ||(W_V)_{:,j}||_2^2 + o(1), \text{ with:}$$

$$s(\rho) = n \int\limits_{-\infty}^{\infty} \int\limits_{-\infty}^{\infty} \Phi_{\rho^2}(x,y)^{n-1} f_{\rho^2}(x,y) dx dy$$

*where $\Phi_c$ is the joint CDF of a bivariate normal distribution with correlation $c$ and $f_c$ is the respective PDF.*

*Proof.* We first prove that each row of the softmax matrix is a permutation:

**Lemma 10.** *Let $P_{i,j} := x_i^T W x_j$. Then:*

$$\sigma(P_{i,1}, ..., P_{i,n}) \overset{p}{\to} e_k, \text{ where } k = \arg\max_{j \in [n]} P_{i,j}$$

*A similar statement also holds for $Y$.*

*Proof.* The proof of this statement is very similar to our argument from Theorem 5.2. We have that:

$$P_{ij} = \sum_{a,b \in [d]} x_{i,a} W_{a,b} x_{j,b}$$

Conditioning on $X$, $P_{ij}$ is normal with mean $0$ and variance $||x_i||_2^2 \cdot ||x_j||_2^2 = \Theta(d^2)$ (Lemma 8). So we can write $P_{ij} = d \cdot Y_{ij}$ where $Y_{ij} \sim \mathcal{N}(0,1)$ conditioned on $X$. Let $k^* = \arg\max_{j \in [n]} P_{i,j} =$

$\arg\max_{j\in[n]} Y_{i,j}$. Let $\Delta_k = P_{ik^*} - P_{ik} = d(Y_{ik^*} - Y_{ik})$. We know that for $k \neq k^*$ the quantity $\Delta_k$ converges to $\infty$ in probability as $d \to \infty$, so:

$$\frac{e^{P_{ik}}}{\sum_s e^{P_{is}}} = \frac{e^{-\Delta_k}}{1 + \sum_{s\neq k^*} e^{-\Delta_s}} \xrightarrow{p} 0$$

as $d \to \infty$. For $k = k^*$ however, we have:

$$\frac{e^{P_{ik^*}}}{\sum_s e^{P_{is}}} = \frac{1}{1 + \sum_{s\neq k^*} e^{-\Delta_s}} \xrightarrow{p} 1$$

This establishes the lemma. $\qquad\square$

Now, let $L_{ij} = y_i^T W y_j$ and let $s = \Pr[\arg\max_{j\in[n]} P_{i,j} = \arg\max_{j\in[n]} L_{i,j}]$. We know that $P_{ij}$ and $L_{ij}$ have correlation $\rho^2$ as:

$$\mathbb{E}\left[\left(\sum_{a,b\in[d]} x_{i,a} W_{a,b} x_{j,b}\right) \cdot \left(\sum_{a,b\in[d]} y_{i,a} W_{a,b} y_{j,b}\right)\right] = \sum_{a,b,c,d\in[n]} \mathbb{E}[x_{i,a} x_{j,b} y_{i,c} y_{j,c} W_{a,b} W_{c,d}] = d\rho^2$$

The joint conditional distribution of both $P_{i,j}$ and $L_{i,j}$ is a standard bivariate normal with correlation $\rho^2$:

$$f_{\rho^2}(x,y) = \frac{1}{2\pi\sqrt{1-\rho^4}} \exp\left(-\frac{x^2 - 2\rho^2 xy + y^2}{2(1-\rho^4)}\right)$$

It's CDF is:

$$\Phi_{\rho^2}(x,y) = \int_{-\infty}^{x} \int_{-\infty}^{y} f_{\rho^2}(u,v)\, du\, dv$$

Thus, we can calculate:

$$s = s(\rho) = n \int_{-\infty}^{\infty} \int_{-\infty}^{\infty} \Phi_{\rho^2}(x,y)^{n-1} f_{\rho^2}(x,y) dx dy$$

The remainder of the proof concludes as in the main text by considering the events of the maxima matching or not, taking the expectation and de-conditioning. $\qquad\square$

## H  FULL NOISE STABILITY DAMPENING IN MULTI-LAYER TRANSFORMERS

Let $W_Q W_K = I$ and $||(W_V)_{:,j}||_2 = \gamma \leq 1$. Ignoring distributional shifts, we can combine Theorem 5.1 and Theorem 5.2 to get the following recurrence:

$$\rho_L = \frac{1}{2\pi}\left(\sqrt{1 - \gamma^4 \rho_{L-1}^2} + \gamma^2 \rho_{L-1}(\pi - \arccos(\gamma^2 \rho_{L-1}))\right) \tag{8}$$

Substituting the linear approximation of Equation (4) and setting $\rho_1 = \frac{1}{2}$, we obtain:

$$\rho_L = \frac{1}{2\pi} + \frac{\gamma^2}{4}\rho_{L-1} \implies \rho_L = \frac{2}{\pi(4-\gamma^2)} + \left(\frac{1}{2} - \frac{2}{\pi(4-\gamma^2)}\right) \cdot \left(\frac{\gamma^2}{4}\right)^{L-1}$$

This suggests that for $\gamma \leq 1$ the noise stability propagation through a multi-layer transformer converges to $\frac{2}{\pi(4-\gamma^2)}$.

However, empirical verification suggests this is not the case. Figure 10 shows that for $\gamma < 1$ we observe full dampening, while for $\gamma = 1$ weak dampening remains. This is due to the fact that for $\gamma \neq 1$ the output of the Transformer layer decreases exponentially with $\gamma$, which also affects the noise stability.

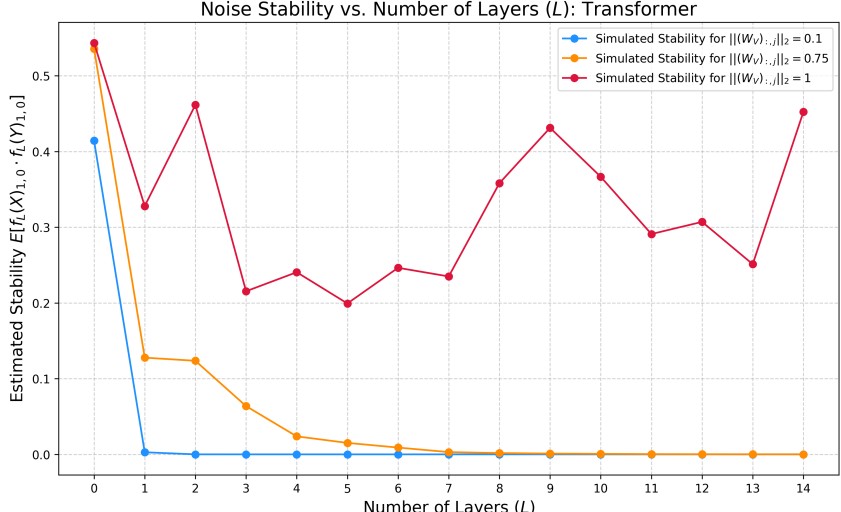

Figure 10: Transformer exhibits full dampening in the multi-layer setting.

# I  NOISE STABILITY INTERVAL PROPAGATION

In this section we present some analytical results on the propagation of noise stability intervals. These are lemmata that provide more flexibility towards the task of analyzing noise stability throughout a deep Transformer, though distributional assumptions still need to be made. Such assumptions can be quite detrimental to the validity of the analysis, as we have already seen.

## I.1  MLP STABILITY UNDER BONAMI-BECKNER GAUSSIANS

**Lemma 11** (Propagation of Stability in MLP Layer). *Let $X \in \mathbb{R}^{n \times d}$ be a random variable where $X_{ij} \sim \mathcal{N}(\mu_{ij}, \sigma_{ij})$ (not necessarily independent of each other). Consider a random variable Y generated by a scaled Bonami-Beckner noise process:*

$$Y_{ij} = \begin{cases} \alpha X_{ij}, & \text{with probability } \rho_{ij} \in [0,1] \\ \sim \mathcal{N}(\mu_{ij}, \sigma_{ij}), & \text{iid, otherwise} \end{cases}$$

*where $\alpha > 0$. Let $\phi : \mathbb{R} \to \mathbb{R}$ be the element-wise ReLU function. Then we have that:*

$$\mathbb{E}[\phi(X_{ij}) \cdot \phi(Y_{ij})] = \rho_{ij}\alpha E_1 + (1 - \rho_{ij})E_2^2$$

*for all $(i,j) \in [n] \times [d]$, where:*

$$E_1 = \sigma_{ij} f\left(\frac{\mu_{ij}}{\sigma_{ij}}\right) + \mu_{ij}\Phi\left(\frac{\mu_{ij}}{\sigma_{ij}}\right)$$

$$E_2 = (\mu_{ij}^2 + \sigma_{ij}^2)\Phi\left(\frac{\mu_{ij}}{\sigma_{ij}}\right) + \sigma_{ij}\mu_{ij} f\left(\frac{\mu_{ij}}{\sigma_{ij}}\right)$$

*and $f, \Phi$ is the PDF and CDF of the standard $\mathcal{N}(0,1)$ Gaussian distribution.*

*Proof.* Let us fix some $i, j$ and use the law of total expectation:

$$\mathbb{E}[\phi(X_{ij})\phi(Y_{ij})] = \rho_{ij}\alpha\mathbb{E}[\phi(X_{ij})^2] + (1 - \rho_{ij})\mathbb{E}[\phi(X_{ij})]^2$$

For the second moment, we have:

$$\mathbb{E}[\phi(X_{ij})^2] = \int_0^\infty z^2 \frac{1}{\sqrt{2\pi\sigma_{ij}^2}} e^{-\frac{(z-\mu_{ij})^2}{2\sigma_{ij}^2}}$$

$$= \frac{1}{\sqrt{2\pi}} \int_{-\mu_{ij}/\sigma_{ij}}^\infty (u\sigma_{ij} + \mu_{ij})^2 e^{-\frac{u^2}{2}} du \qquad\qquad (u = \tfrac{z-\mu_{ij}}{\sigma_{ij}})$$

$$= \frac{\sigma_{ij}^2}{\sqrt{2\pi}} \int_{-\mu_{ij}/\sigma_{ij}}^\infty u^2 e^{-\frac{u^2}{2}} du + \frac{2\sigma_{ij}\mu_{ij}}{\sqrt{2\pi}} \int_{-\mu_{ij}/\sigma_{ij}}^\infty u e^{-\frac{u^2}{2}} du + \frac{\mu_{ij}^2}{\sqrt{2\pi}} \int_{-\mu_{ij}/\sigma_{ij}}^\infty e^{-\frac{u^2}{2}} du$$

For the third summand in this expression, we have that:

$$\frac{\mu_{ij}^2}{\sqrt{2\pi}} \int_{-\mu_{ij}/\sigma_{ij}}^\infty e^{-\frac{u^2}{2}} du = \mu_{ij}^2 \left(1 - \Phi\left(-\frac{\mu_{ij}}{\sigma_{ij}}\right)\right) = \mu_{ij}^2 \Phi\left(\frac{\mu_{ij}}{\sigma_{ij}}\right)$$

For the second summand, it is:

$$\frac{2\sigma_{ij}\mu_{ij}}{\sqrt{2\pi}} \int_{-\mu_{ij}/\sigma_{ij}}^\infty u e^{-\frac{u^2}{2}} du = \frac{2\sigma_{ij}\mu_{ij}}{\sqrt{2\pi}} \left[-e^{u^2/2}\right]_{-\mu_{ij}/\sigma_{ij}}^\infty = 2\mu_{ij}\sigma_{ij} f\left(\frac{\mu_j}{\sigma_{ij}}\right)$$

For the first summand, we can use integration by parts to get:

$$\frac{\sigma_{ij}^2}{\sqrt{2\pi}} \int_{-\mu_{ij}/\sigma_{ij}}^\infty u^2 e^{-\frac{u^2}{2}} du = \frac{\sigma_{ij}^2}{\sqrt{2\pi}} \cdot \left(\left[-u e^{-u^2/2}\right]_{-\mu_{ij}/\sigma_{ij}}^\infty + \int_{-\mu_{ij}/\sigma_{ij}}^\infty e^{-\frac{u^2}{2}} du\right)$$

$$= \sigma_{ij}^2 \frac{-\mu_{ij}}{\sigma_{ij}} f\left(\frac{\mu_{ij}}{\sigma_{ij}}\right) + \sigma_{ij}^2 \Phi\left(\frac{\mu_{ij}}{\sigma_{ij}}\right)$$

Combining, we get an expression for the second moment:

$$E_1 := \mathbb{E}[\phi(X_{ij})^2] = \sigma_{ij}^2 \frac{-\mu_{ij}}{\sigma_{ij}} f\left(\frac{\mu_{ij}}{\sigma_{ij}}\right) + \sigma_{ij}^2 \Phi\left(\frac{\mu_{ij}}{\sigma_{ij}}\right) + 2\mu_{ij}\sigma_{ij} f\left(\frac{\mu_j}{\sigma_{ij}}\right) + \mu_{ij}^2 \Phi\left(\frac{\mu_{ij}}{\sigma_{ij}}\right)$$

$$= (\sigma_{ij}^2 + \mu_{ij}^2)\Phi\left(\frac{\mu_{ij}}{\sigma_{ij}}\right) + \mu_{ij}\sigma_{ij} f\left(\frac{\mu_j}{\sigma_{ij}}\right)$$

Now for the mean, we have:

$$\mathbb{E}[\phi(X_{ij})] = \int_0^\infty z \frac{1}{\sqrt{2\pi\sigma_{ij}^2}} e^{-\frac{(z-\mu_{ij})^2}{2\sigma_{ij}^2}}$$

$$= \frac{1}{\sqrt{2\pi}} \int_{-\mu_{ij}/\sigma_{ij}}^\infty (u\sigma_{ij} + \mu_{ij}) e^{-\frac{u^2}{2}} du \qquad\qquad (u = \tfrac{z-\mu_{ij}}{\sigma_{ij}})$$

Splitting up this sum, we have:

$$E_2 := \mathbb{E}[\phi(X_{ij})] = \frac{\sigma_{ij}}{\sqrt{2\pi}} \int_{-\mu_{ij}/\sigma_{ij}}^\infty u e^{-\frac{u^2}{2}} du + \frac{\mu_{ij}}{\sqrt{2\pi}} \int_{-\mu_{ij}/\sigma_{ij}}^\infty e^{-\frac{u^2}{2}} du$$

$$= \sigma_{ij} f\left(\frac{\mu_{ij}}{\sigma_{ij}}\right) + \mu_{ij} \Phi\left(\frac{\mu_{ij}}{\sigma_{ij}}\right)$$

This concludes the proof. $\qquad\square$

## I.2 STABILITY PROPAGATION THROUGH A SINGLE ATTENTION LAYER

For convenience, the following result defines a **stability matrix**, which captures the correlation of each input token with the other. Assuming all the entries in that matrix are bounded in some interval, we analyze the noise stability propagation through an attention layer. We also assume some structural properties of the weight matrices to allow for stability propagation in our proof:

**Lemma 12** (Stability Propagation through an Attention Layer). *Let $X, Y$ have stability matrix $\{C\}_{k\ell,k'\ell'} = \mathbb{E}[X_{k\ell}Y_{k'\ell'}] \in \mathbb{R}^{nd \times nd}$ with $0 < \rho_\ell \leq C_{k\ell,k'\ell'} \leq \rho_r$ for all $k, k' \in [n], \ell, \ell' \in [d]$, and suppose $||X||_\infty, ||Y||_\infty \leq B$ with probability 1. Consider an attention layer with matrices $W_K, W_Q, W_V \in \mathbb{R}^{d \times d}$ and denote, for two vectors $x, y \in \mathbb{R}^d$ the following quantities:*

$$S^+(x,y) = \sum_{i,j \in [d]} \max(0, x_i y_j), \quad S^-(x,y) = \sum_{i,j \in [d]} \min(0, x_i y_j)$$

*Suppose that $W_V$ is such that $\rho_\ell S^+(w_j, w_{j'}) + \rho_r S^-(w_j, w_{j'}) > 0$ for all $j, j' \in [d]$, where $w_j = (W_V)_{:,j}$ is the $j$-th column of $W_V$. Hence, let:*

$$R_\ell := \inf_{j,j' \in [d]} \{\rho_\ell S^+(w_j, w_{j'}) + \rho_r S^-(w_j, w_{j'})\} > 0$$

$$R_r := \sup_{j,j' \in [d]} \{\rho_r S^+(w_j, w_{j'})\}$$

*Now, let $A(X) \in \mathbb{R}^{n \times d}$ be the output of the attention layer. Then we have that:*

$$0 < \frac{R_\ell}{E} \leq \mathbb{E}(A(X)_{ij} A(Y)_{i'j'}) \leq R_r \cdot E \tag{9}$$

*for all $i, i' \in [n], j, j' \in [d]$, where $E := \exp(4d^2 B^2 ||W_K||_\infty ||W_Q||_\infty)$.*

*Proof.* Let $S^X = \sigma(XW_Q W_K^T X^T)$ and $V^X := XW_V$. We have that $A(X)_{ij} = \langle S_{i,:}^X, V_{:,j}^X \rangle$, so:

$$
\begin{aligned}
\mathbb{E}(A(X)_{ij} \cdot A(Y)_{i'j'}) &= \mathbb{E}\left[\langle S_{i,:}^X, V_{:,j}^X \rangle \cdot \langle S_{i,:}^Y, V_{:,j}^Y \rangle\right] \\
&= \mathbb{E}\left[\left(\sum_{k=1}^n S_{ik}^X V_{kj}^X\right) \cdot \left(\sum_{k'=1}^n S_{i'k'}^Y V_{k'j}^Y\right)\right] \\
&= \mathbb{E}\left[\sum_{k,k'} S_{ik}^X S_{i';k'}^Y V_{kj}^X V_{k'j'}^Y\right]
\end{aligned}
$$

Now let $q_i^X := X_{i,:} \cdot W_Q \in \mathbb{R}^d$, $k_i^X = X_{i,:} \cdot W_K$. We have:

$$S_{ik}^X = \frac{\exp(q_i^X \cdot k_k)}{\sum_{s=1}^n \exp(q_i^X \cdot k_s)}$$

Therefore:

$$
\mathbb{E}(A(X)_{ij} \cdot A(Y)_{i'j'})
$$

$$
= \mathbb{E}\left[\sum_{k,k'} \frac{\exp(q_i^X k_k)}{\sum_{s=1}^n \exp(q_i^X k_s)} \cdot \frac{\exp(q_{i'}^Y k_{k'})}{\sum_{s=1}^n \exp(q_{i'}^Y k_s)} \cdot V_{kj}^X V_{k'j'}^Y\right]
$$

$$
= \mathbb{E}\left[\sum_{k,k'} \frac{\exp(X_{i,:} W_Q W_K^T X_{k,:}^T)}{\sum_{s=1}^n \exp(X_{i,:} W_Q W_K^T X_{s,:}^T)} \cdot \frac{\exp(Y_{i',:} W_Q W_K^T Y_{k',:}^T)}{\sum_{s=1}^n \exp(Y_{i',:} W_Q W_K^T Y_{s,:}^T)} \cdot X_{k,:}(W_V)_{:,j} Y_{k',:}(W_V)_{:,j'}\right]
$$

We will bound the softmax terms using the norms of $W_Q, W_K$ and $X$. For any $s_1, s_2$ we have:

$$|X_{s_1,:} W_Q W_K^T X_{s_2,:}^T| \leq d^2 B^2 ||W_Q||_\infty \cdot ||W_K||_\infty$$

And this implies that:

$$\frac{1}{n}\exp(-2d^2 B^2 ||W_K||_\infty ||W_Q||_\infty) \leq \frac{\exp(X_{i,:} W_Q W_K^T X_{k,:}^T)}{\sum_{s=1}^n \exp(X_{i,:} W_Q W_K^T X_{s,:}^T)} \leq \frac{1}{n}\exp(2d^2 B^2 ||W_K||_\infty ||W_Q||_\infty)$$

We now analyze the isolated terms

$$\mathbb{E}[X_{k,:}(W_V)_{:,j}Y_{k',:}(W_V)_{:,j}]$$

We can use our prior information on the stability matrix $C$. Fix some $k, k'$ and we have that:

$$
\begin{aligned}
\mathbb{E}[X_{k,:}(W_V)_{:,j}Y_{k',:}(W_V)_{:,j}] &= \sum_{\ell,\ell'} \mathbb{E}\left[X_{k\ell}(W_V)_{\ell j}Y_{k'\ell'}(W_V)_{\ell'j'}\right] \\
&= \sum_{\ell,\ell'} (W_V)_{\ell j}(W_V)_{\ell'j'} \cdot \mathbb{E}\left[X_{k\ell}Y_{k'\ell'}\right] \\
&\in [R_\ell, R_r] \qquad\qquad (\text{as } \mathbb{E}\left[X_{k\ell}Y_{k'\ell'}\right] \in (r_\ell, r_r))
\end{aligned}
$$

Overall, if we let $E := \exp(4d^2B^2||W_K||_\infty||W_Q||_\infty)$, we get that:

$$0 < \frac{R_\ell}{E} \leq \mathbb{E}(A(X)_{ij} \cdot A(Y)_{i'j'}) \leq R_r \cdot E$$

$\square$

## J    Noise Stability Regularization Experiments

### J.1    Architecture Layout, Hyperparameters and Training Details

We present details of our architecture, training and hyperparameters. These can also be found in our codebase. Each Transformer layer uses multi-head self-attention followed by a position-wise feed-forward network, with residual connections around both sublayers. We use sinusoidal positional encodings, as well as binary attention masking $M$. We also use dropout, applied to attention weights, attention output, and FFN hidden layer with rate $p$. The activation function we use for our FFN is ReLU. To produce a classification label we use mean pooling over the label dimension.

For initialization, linear layers are initialized with $\mathcal{N}(0, 0.02^2)$ and zero biases. Every other learnable parameter is initialized via Xavier initialization. Positional encodings are fixed and not trainable.

For training, we use AdamW with learning rate $\eta$ and $\ell_2$ weight decay regularization $\lambda$. Our loss is the cross entropy loss. We use a learning rate scheduler that reduces $\eta$ on validation loss plateau. The patience and factor parameters of the scheduler are hyperparameters we set. Our codebase also provides support for multi-GPU and distributed training, though our models were too small to benefit from such augmentations.

Table 2: Model Hyperparameters and Training Configuration for Modular Addition

| Name | Symbol | Default |
|---|---|---|
| Embedding dimension | $d_{\text{model}}$ | 128 |
| Transformer layers | $L$ | 2 |
| Attention heads | $H$ | 2 |
| Max seq length (PE) | `max_length` | 512 |
| Vocab size | $|\mathcal{V}|$ | $K + 5$ |
| Num classes | $C$ | $K = 113$ |
| Dropout rate | $p$ | 0.1 |
| Batch size | $B$ | 256 |
| Epochs | $T$ | 7000 |
| Learning rate | $\eta$ | 0.001 |
| Weight decay | $\lambda$ | 0.001 |
| Label smoothing | – | 0.0 |
| Scheduler patience | – | 10 epochs |
| Scheduler factor | – | 0.8 |
| Train samples | – | 2000 |

Continued on next page

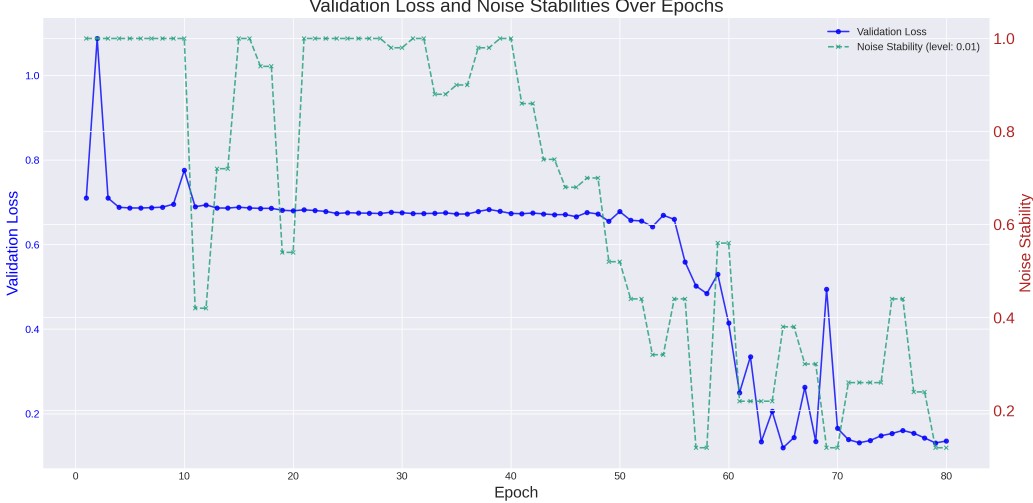

Figure 11: Evolution of noise stability and validation loss for noisy sparse parity. Stability (*green*) begins to decrease before the validation loss (*blue*) drops, acting as a leading indicator for generalization.

| Table 2 – continued from previous page | | |
| --- | --- | --- |
| **Name** | **Symbol** | **Default** |
| Val/Test samples | – | 200 / 200 |

## J.2 Evolution of Noise Stability During Training.

We also analyze how noise stability (with $\rho = 1/2$) evolves during training without regularization, focusing on the noisy sparse parity (NSP) task first because its target function has a known, low noise stability of $\rho^k$ (O'Donnell, 2021). We make three key observations:

- A randomly initialized Transformer exhibits high noise stability, which decreases throughout training to converge toward the low stability level of the target function (Figure 11).

- Noise stability is a leading indicator for generalization. Figure 11 shows that stability begins to decrease well before the sharp drop in validation loss, signaling that internal model adjustments precede performance improvements.

- Stability can serve as a secondary metric for model selection. Among models with similar validation loss, the one whose stability best aligns with the theoretical properties of the target function may be preferable.

## J.3 Experiments on Language Generation

We also tested noise stability regularization on a language generation task. We trained a 4 layer transformer model with $d_{\text{model}} = 30$ and $H = 6$ on the next-token-prediction task. The dataset we used was *WikiText-2-v1 (Small)*, with sequence length $N = 20$, a vocabulary size of 500, batch size of 200 and 1000 training examples[8]. We trained our transformer without noise stability regularization, with weight decay $\lambda = 0.02$ and with numerous settings of $(\rho, \gamma)$. We tracked noise stability, validation loss and validation accuracy throughout training.

We first observe that even in this non-synthetic task noise stability regularization offers deep benefits for training. As shown in Figure 12, the model climbs to $70\%$ accuracy within 1000 iterations in the grokking phase. On the other hand, it takes the non-regularized model 4000 iterations to do

---

[8]Our setup can also be found in full in our codebase.

so, amounting to a 75% speedup. The situation is similar for the validation losses: the regularized models exhibit more stability in training, while the non-regularized training fluctuates in validation loss. Noise stability regularization also causes the validation loss to decrease much faster and earlier than without it, while grokking.

Examining the noise stability at $\rho = 1/2$ for the regularized and non-regularized settings (Figure 14) we can see a fundamental difference between the models. The non-regularized model becomes less and less stable, which could explain its instability. Regularized models stay stable as the training dynamics force the model to improve while remaining robust. Understanding the benefits of this regularization better is definitely an interesting direction for future work.

Finally, we can see that noise stability regularization offers benefits for a variety of different settings of the hyperparameters $(\rho, \gamma)$. Ultimately however, the best performance is found via a thorough hyperparameter sweep, as Figure 13 shows.

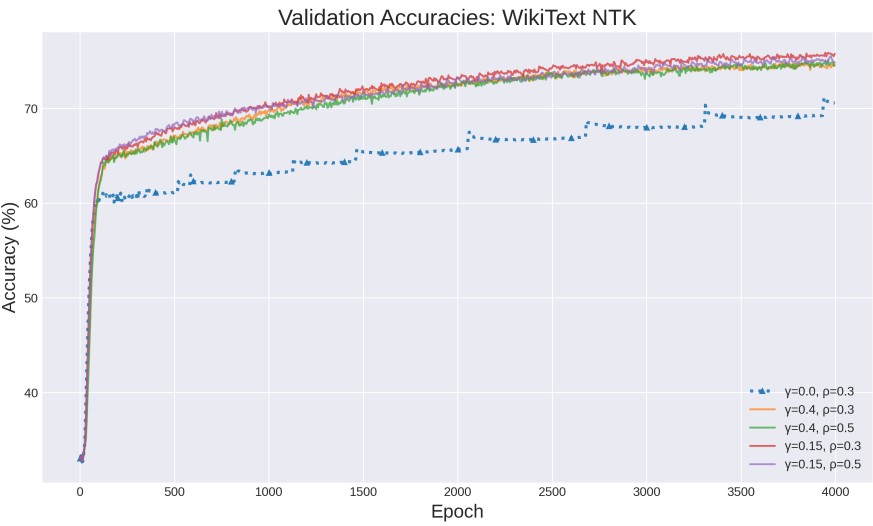

Figure 12: Accuracy Comparison on Next-Token-Prediction

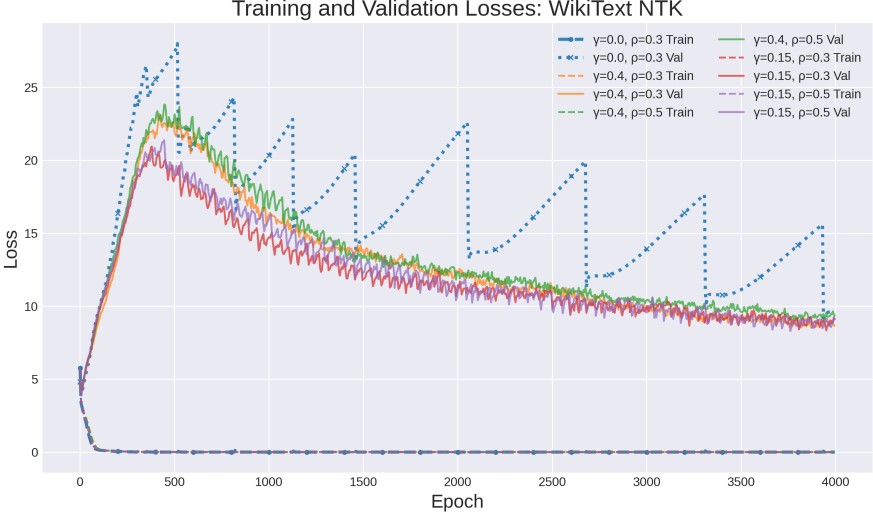

Figure 13: Training and Validation Losses for Next-Token Prediction

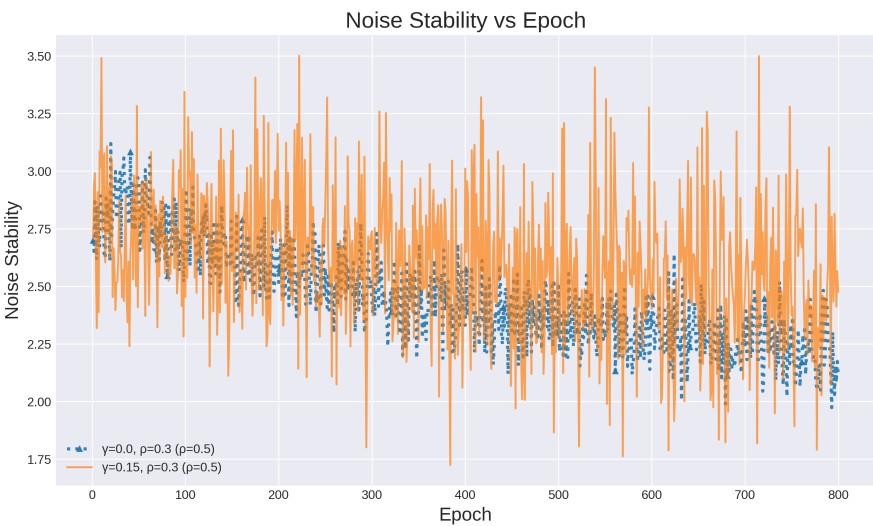

Figure 14: Noise Stability Comparison: Regularization vs Non-Regularization. We see that the non-regularized model tends to become more unstable, while regularization maintains stability.

