# OpenReview forum: "Noise Stability of Transformer Models"
_ICLR.cc/2026/Conference — ICLR 2026 Poster_

### Official Review · Reviewer_6UoK · 2025-10-29

**Soundness:** 3
**Presentation:** 3
**Contribution:** 3
**Rating:** 8
**Confidence:** 4

**Summary:**

The authors propose a method to investigate the simplicity bias in neural networks. Simplicity bias is the tendency of neural networks of learning simple functions to fit the data: as simpler functions are (i) more interpretable and (ii) more robust, assessing simplicity is important to study model reliability.
A current trend of research frames the problem of simplicity in terms of spectral decomposition. The underlying idea is that simple functions should show spectral concentration around low-order polynomials. Drawing from Boolean functional analysis, it is possible to study the impact of a single token by "swapping" its sign and contrasting the average output change with the total variance. The authors criticise the approach. They claim that neural networks employ functions defined on real numbers -and not on binary coordinates- and they criticise the idea of considering one token at the time as this fails to capture their collective behaviour: (i) the "junta"-like concentration, i.e., the sparsity of the influential tokens; (ii) the positional bias; (iii) the non-zero influence of all tokens.
To address their concern, they propose to extend the concept of noise stability from Boolean analysis - which is the study of the effect of correlated noise applied to all inputs simultaneously. First, they define the concept of stability as the correlation between the original input and a partially perturbed input - corrupted with Gaussian noise. Then, they express this measure employing the spectral decomposition of the function in terms of its Hermite-Fourier coefficients. Finally, they study (i) a simplified RELU MLP and (ii) a simplified Transformer, finding that the first dampens stability, whereas the latter maintains it.
To test the impact of penalising with respect to spectral concentration, they propose a stability regularisation: they train transformers on two artificial tasks that exhibit grokking, i.e., a sharp decrease in the loss function value after an initial over-fitting phase, noting that regularisation cuts the number of iterations requires to attain it.

**Strengths:**

Originality:  The authors propose an approach to the study of neural networks that is well-grounded in theory and takes into account the collective behaviour of the tokens in LLMs. They derive analytical solutions for typified examples and introduce a regularisation mechanism.
 Quality:   the derivations are sound; experiments accompany all the claims; a numerical example works as a case study.
  Clarity: The work is clear.  well-organised and easy to read. The Appendix is organised well and it is easy to follow the derivation of the derivation of the Ornstein-Uhlenbeck Operator. Some relevant entries in the bibliography are incomplete (e.g., Nathan Keller. Elchanan Mossel. Arnab Sen. "Geometric influences." Ann. Probab. 40 (3) 1135 - 1166, May 2012. https://doi.org/10.1214/11-AOP643). The labels of the plots are too tiny and difficult to read.
 Significance The author propose a new method to investigate the simplicity bias that considers all tokens at the same time. Given the importance of collective token dynamics in Transformer, this work goes in the right direction.

**Weaknesses:**

The model criticizes the use of influence as a proxy of sensitivity and propose noise stability as a new method. One of the reasons of their criticism is that the upper bound of the former method are too loose with respect to the latter when estimating spectral concentration. As evidence, they report the values for four different models in Table 1. However, it is not entirely clear if this is the only advantage of their new method. In other words, it is not entirely clear if using influence or, better still, geometric influence it is not possible to gain the insights they obtain using noise stability. The authors obtain interesting results for MLPs and Transformers, but it is not  clear why the results differ and what might be the reason behind this. Similarly, while the example of Section 6 shows that there seems to a be link between simplicity bias and spectral concentration, it is not immediately clear if noise stability captures it better than influence, if it might be possible to use it too, and if there is a way to experimentally detect the issues they already showcase in Figure 1 in this training setting.

**Questions:**

1. Line 107/108. Are you sure the dimensions of this equation are correct?
    2. Table 1. What would happen if we instead considered the geometric influence here?
    3. Line 279. Maybe a typo: you see "(see Section 5.2)" in Section 5.2.
    4. Figure 2. This is a bit hard to read: the labels are very small.
    5. Section 5. Can you comment on the results? Why are the different between Transformers and MLPs? Or how does it show in practice?

---

> ### Author Response · Authors · 2025-11-16
> **Discussion Comments for Reviewer 6UoK**
>
> We thank the reviewer for their thoughtful comments and insights.
>
> 1. **On influence vs noise stability**: As the reviewer noted, noise stability as a metric is not necessarily comparable with influence. Both of these metrics give helpful insights about functions and they both have valuable correlations with interesting neural network properties. We think that combining these properties and analyzing them further is an interesting future direction.
> 2. **Transformers vs MLPs:** Self-attention definitely causes a drastic rift between vanilla neural networks and Transformers, especially in the context of moment or noise stability propagation. Though we identify and exhibit this phenomenon in theory and practice, we do not resolve this picture fully. We believe this is a valuable and interesting open question.
> 3. **Question 1:**
>     1. Line 107/108: Yes, each weight matrix for each head is $d\times d$ and the output of the attention function is $\mathbb{R}^{n\times d}$.
>     2. Table 1: We actually computed the geometric influence with respect to the uniform measure here. However, it is a good question whether the measure $\mu$ we use can be changed to explain the concentration. We can definitely consider this as a future direction!
>     3. Line 279: Yes, this refers to “Figure 2”. We will fix the typo.
>     4. Figure 2: We will fix the figure so it is easier to read! Thank you for pointing it out!
>     5. Section 5: We believe that MLPs are likely easier to study as even our proxy recursion allows for valid analysis. Once attention is inserted, the dynamics our results describe scramble the input distribution considerably. This might suggest that a mechanistic analysis should come before a statistical one in practice. In general, when studying moment and distribution propagation throughout Transformers we will need new techniques, tools and insights. We take one small step towards that theoretical direction by proposing stability interval propagation, but more work is required to obtain the full picture here.
>
> We thank the reviewer again for their time and useful comments! Please let us know if you have any further questions.

---

> > ### Comment · Reviewer_6UoK · 2025-11-20
> >
> > Thanks for the replies to my questions and for the clarifications

---

### Official Review · Reviewer_QkLJ · 2025-10-29

**Soundness:** 3
**Presentation:** 2
**Contribution:** 2
**Rating:** 2
**Confidence:** 4

**Summary:**

The paper proposes noise stability as a measure of simplicity of Transformers.
Unlike average sensitivity, which captures local perturbations, noise stability measures robustness to correlated, global perturbations across the input sequence. Building on this idea, the authors introduce a noise-stability regularizer and show that it improves training and generalization on algorithmic tasks exhibiting grokking dynamics.

**Strengths:**

1. The mathematical formulation of noise stability through the Ornstein–Uhlenbeck operator is well presented. The concept of analyzing robustness under global, correlated perturbations rather than local ones is intuitive and sound.
2. The introduction of a differentiable noise-stability regularizer is a concrete and practical contribution.
3. The choice of algorithmic tasks is appropriate for studying grokking phenomena. The experiments show correlations between noise stability, validation loss, and generalization behavior.

**Weaknesses:**

1. The paper motivates the use of noise stability over average sensitivity based on the observed “junta-like” dependence in Transformers, arguing that noise stability captures global correlated perturbations rather than single-coordinate ones. However, the connection between robustness to global perturbations and dependence on a few input variables is not clearly established. It remains unclear why capturing correlated noise should correspond to fewer influential tokens.
2. The authors claim that noise stability provides tighter spectral tail bounds than average sensitivity, but no theoretical comparison or proof is given. The empirical evidence in Table 1 lacks sufficient methodological detail for verification—for example, the meaning of “percentage of weight in degrees ≥ 15” is not explained.
3. The claim that “average sensitivity is too loose to explain the behavior of modern Transformers” is not convincingly supported by experiments. The paper would be much stronger with a direct comparison of average sensitivity and noise stability as regularizers.
4. The paper also does not compare against other regularizers known to affect grokking dynamics (e.g., weight decay, $L_1$ norm [1, 2]). The reported grokking acceleration appears to be based on a limited number of runs and lacks confidence intervals.
5. The proposed regularizer perturbs discrete token inputs via random replacements, whereas the theoretical formulation of noise stability is defined over Gaussian perturbations. The paper would benefit from clarifying how these two settings relate.
6. Section 5’s analysis of noise stability in Transformers does not meaningfully yield actionable insights for Transformer training or regularizer design.

Minor:
- The term “junta-like” appears in the abstract before being defined.
- In Line 397, the authors mention “three key properties” but list only two.

**Questions:**

1. How do the positional-bias and sensitivity patterns in Figure 1 concretely demonstrate the limitation of average sensitivity, and in what way does noise stability resolve it?
2. Can the authors include a controlled comparison among (i) average-sensitivity regularization, (ii) noise-stability regularization, (iii) weight decay, and (iv) no regularization, to demonstrate that noise stability improves training dynamics beyond heuristic smoothing?

3. What is the additional computational overhead of computing and back-propagating the noise-stability term compared to average sensitivity?

4. In Appendix J.2, the authors show that noise stability correlates with validation loss. Can they further test whether noise stability decreases before validation loss changes, serving as an early indicator of grokking as observed in [3].

[1] Ziming Liu, Eric J Michaud, and Max Tegmark. Omnigrok: Grokking Beyond Algorithmic Data. ICLR 2023.

[2] Kaifeng Lyu, Jikai Jin, Zhiyuan Li, Simon S. Du, Jason D. Lee, and Wei Hu. Dichotomy of Early and Late Phase Implicit Biases Can Provably Induce Grokking. ICLR 2024.

[3] Bhavya Vasudeva, Deqing Fu, Tianyi Zhou, Elliott Kau, Youqi Huang, and Vatsal Sharan. Transformers Learn Low Sensitivity Functions: Investigations and Implications. ICLR 2025.

---

> ### Author Response · Authors · 2025-11-16
> **Discussion Comments for Reviewer QkLJ**
>
> We thank the reviewer for their insightful review and questions. We try to address their concerns below:
>
> 1. **Robustness and Global Pertrubations:**
>     1. We adopt a fairly standard definition for robustness: changes to model outputs given small perturbations to inputs. This is captured both by average sensitivity and noise stability in different ways.
>     2. *“Why does capturing correlated noise correspond to fewer influential tokens”?*
>         1. This is where the connection between stability and spectral tails becomes relevant. Theoretically, Lemma 1 explains this phenomenon: a lower bound to noise stability implies an upper bound to the spectral tail. Average Sensitivity's spectral concentration guarantees are given by Friedgut's Junta Theorem, while Lemma 1 describes the guarantees for noise stability. As we discuss below, we observe that the former is too loose for real LLMs and propose noise stability as a tighter alternative.
>         2. Table 1 shows that this spectral concentration is explained better by stability rather than average sensitivity: we measure the weight of the spectral tail by examining all the degrees $d \geq 15$. This is an empirical proof-of-concept evaluation: the top $15$ Fourier coefficients contain a lot more mass under the noise stability interpretation, which suggests higher concentration.
> 2. **Comparing Average Sensitivity and Noise Stability as Regularizers**
>     1. *(Also **answering Q1**)* Our claim towards the looseness of average sensitivity is supported mainly by the experiment of Figure 1, where we show, on numerous real LMs that the number of influential coordinates is much smaller than what average sensitivity predicts: Friedgut's junta theorem suggests that the number of variables with influence at least $0.1 I[f]$ is at least $2^{10} = 1024$. But we observe only $5$-$10$ coordinates. This suggests a “looseness” in the bound. We acknowledge that there might be some confusion in our description, so we updated our manuscript to increase clarity in this point.
>     2. It is an interesting question whether average sensitivity can be utilized as a regularizer to accelerate grokking. It is known from prior work [Vasudeva et al; 2025] that it at least serves as an indicator for grokking, but in our experiments we did notice that it can catalyze it. We leave a more extensive comparison of the two regularization methods as future work.
> 3.  **Confidence in regularizer effects**
>     1. As we mention in Section 6, noise stability regularization is applied *on top* of other regularization methods (eg. weight-decay) and it still provides the benefits of faster grokking.
>     2. **Question 2**: We ran our experiments for a wide variety of inputs and random seeds for the synthetic problems presented (see attached codebase). To inspire further confidence, and in line with other reviewers’ suggestions, we also ran experiments on a next-token-prediction task on WikiText-v2. Our findings support the conclusions we drew from our synthetic experiments and updated our manuscript to include them (see updated Section 6.1):
>    * Under noise stability regularization the validation loss and perplexity diminish at the grokking stage much faster than without regularization.
>    * The regularized model achieves higher accuracy than the non-regularized one while only taking 25% as many iterations.
>    * Inline with theory and intuition, training with noise stability regularization leads to more stable model, whereas grokking without regularization causes the stability to decrease. This means that it is highly unlikely our regularization constitutes a heuristic smoothing (**Q2**). As an **answer to *Q4***, we mention in Appendix J2 that indeed noise stability drops before validation loss does, which means it indeed indicates grokking.
> 4. **On the definition of the regularizer:**
>     *  We use the uniform distribution in practice because our token space is discrete. Apart from this difference the definitions are morally identical.
> 5. **Actionable Insights from our Theoretical Contributions**: Our analysis in Sections 4 and 5 aim to set foundational tools for theoretically studying stability in Transformers. Such results can lead to important insights towards explaining transformer behavior.
> 6. **Answering Q3**: This is also a very good and important point and we thank the reviewer for bringing it up. Computing noise stability is no harder than average sensitivity: it takes a constant number of additional forward passes through the network. While not trivial, this is a very scalable cost that could be integrated into training given the benefits of our regularizer. Developing more cost efficient ways to estimate noise stability in a network is a fascinating open problem. We will add a brief discussion of this topic in our manuscript.
>
> Again, we appreciate the reviewer’s help and time. Please let us know if you have any additional questions.

---

> > ### Comment · Reviewer_QkLJ · 2025-11-27
> >
> > I thank the authors for clarifying the motivation behind using Noise Stability over Average Sensitivity. While the explanation addressed the loose bounds of Average Sensitivity, it remains unclear how Noise Stability specifically helps address or explain 'positional bias' and 'sensitivity.'
> >
> > Could the authors further clarify what specific insights the analysis in Section 5 offers regarding Transformer behavior?
> >
> > Additionally, the results in Section 6.1 are currently too concise to be fully understood without referencing Appendix J.3. Given the importance of this experiment, I suggest moving key experimental details into the main text to improve readability.
> >
> > Most importantly, I believe the paper would benefit significantly from a direct comparison of Average Sensitivity and Noise Stability as regularizers, as this aligns directly with the paper's motivation. Therefore, I strongly advise including this comparison in future revisions.

---

> ### Author Response · Authors · 2025-12-01
>
> We thank the reviewer for their response!
>
> * Noise stability and average sensitivity address *simplicity bias* in the sense that the model's output is disproportionately influenced by a small number of coordinates. Our theoretical exposition adopts and builds upon that perspective. Perhaps the reviewer refers to a different notion of "positional bias" here, which our definitions do not capture. Notions inspired from boolean function analysis do not give more insight into why specific token positions are highlighted - they only showcase that few of them are.
> * We agree that Section 5 could benefit from concrete, extractable insights of the practical Transformer architecture. Our intention with that section was primarily to perform a theoretical analysis using Ornstein-Uhlenbeck operator theory. That being said, we are able to extract some useful conclusions from our results that we will include in our revision:
>     - We show that stability is directly proportional to the column norms of the $W_V$ matrix. Ensuring these norms do not diminish during training is a practical guideline suggested by this theorem. Techniques such as spectral normalization and weight decay could be used to achieve, among others, exactly this outcome.
>     - We show that MLPs maintain stability throughout multiple layers while attention layers diminish it when $W_V$ has norms close to $0$. To maintain stability over multiple layers techniques like spectral normalization of $W_V$ could be useful.
> * We agree that Section 6.1 deserves more real estate into the main paper. We will shift some parts from Appendix H.3 to that section in our revision.
> * **On comparing average sensitivity regularization and noise stability regularization**: We agree this experiment would be an impactful addition to our work. With the synthetic experiments of addition and parity we found that average sensitivity regularization did not exhibit the grokking effect that noise stability regularization does. However, when running the language generation task on wikitext-v2 with average sensitivity regularization we actually do observe that grokking is accelerated, albeit in a different way it is with noise stability regularization: while the former leads validation accuracy to increase faster, the latter leads validation loss to decrease faster. Noise stability regularization reaches a lower validation loss in the same time eventually and results in a better model, though they both outperform the absence of regularization:
>
> | Regularization Method | Iterations to 55% Accuracy | Iterations to 20 Val Loss | Final Validation Accuracy |
> |------------------------|---------------------------|---------------------------|---------------------------|
> | (a) No Regularization | 4000 | 3333 | 55.56%|
> | (b) Noise Stability Reg | 1051 | 2000 | 63.24%|
> | (c) Avg Sensitivity Reg | 701 | 3700 | 61.87%|
>
> While there is definitely more work required to properly compare the two, we conjecture that the two types of regularization could be suited better for different tasks. Noise stability regularization is a more aggressive regularizer the penalizes the entire sequence, while average sensitivity only changes one token index. Some tasks could benefit more from the latter, and we believe that exploring this phenomenon in full is a very compelling direction for future work. We will include a discussion of these results and a comparison between the two regularization methods in our revision.
>
> We thank the reviewer for their engagement and we hope we have addressed their concerns adequately.

---

### Official Review · Reviewer_DkBU · 2025-10-31

**Soundness:** 2
**Presentation:** 2
**Contribution:** 2
**Rating:** 4
**Confidence:** 2

**Summary:**

This paper proposes noise stability as an alternative to average sensitivity for explaining spectral concentration in Transformers. The authors argue that average sensitivity has theoretical limitations in extending to real-valued domains and empirically fails to capture the "junta-like" behavior observed in modern LLMs. They provide theoretical analysis of noise stability for single-layer attention and ReLU MLP components, develop a stability interval propagation technique for multi-layer networks, and introduce a noise stability regularization method that accelerates grokking by approximately 35% on algorithmic tasks.

**Strengths:**

**Substantive theoretical contributions.**
The paper gives an exact, closed-form expression for the noise stability of a ReLU layer (Theorem 5.1 / Appendix E) and analyzes single-layer attention under different structural regimes (identity/low-rank and unstructured; Theorems 5.2 and 5.3). It also studies multi-layer propagation, showing weak dampening for deep ReLU MLPs and contrasting behavior for Transformers.

**Thorough and self-contained presentation.**
The work includes background on Boolean function analysis and the Ornstein–Uhlenbeck semigroup/Hermite expansion, connecting noise stability to spectral concentration and supporting later results with detailed appendices.

**Weaknesses:**

**Limited comparison with prior works.**
Section 3 argues that geometric influence (GI) is a more robust alternative to average sensitivity in continuous domains, but then pivots to advocating noise stability as the primary metric without a direct empirical or theoretical head-to-head comparison (e.g., GI vs. noise stability on the same tasks/models). Related work cites Li & Mossel (2025) for noise sensitivity propagation but does not clearly mention what is new beyond the inspiration or compare results. Adding targeted comparisons would sharpen the contribution.

**The regularizer mechanism requires more intuition and ablation.**
The regularizer (Def. 6.1) is clearly specified, but the paper offers limited intuition for why this particular form steers learning toward “simpler” functions beyond the variance-level view. Hyperparameter sweeps for $\gamma$ and $\rho$ are also missing.

**Narrow experimental scope.**
All experiments are on synthetic algorithmic tasks (noisy k-sparse parity, modular addition), and the grokking speed-up is reported only there. It’s unclear whether the regularizer helps on realistic NLP or larger-scale settings.

**Minor presentation issues.**
There’s a typo (“doamins”) in Section 4, and some figures/tables could use more consistent visual styling and larger font sizes for readability.

**Questions:**

1. What is the intended use case for S=0 (encouraging instability)? Are there regimes (e.g., parity-like targets) where anti-stability regularization could help avoid degenerate shortcuts? The paper only evaluates S=1.

2. Appendix J.2 analyzes stability evolution *without* regularization. Showing the same trajectories *with* the regularizer would more directly support the “catalyst” claim and illuminate the mechanism.

3. What is the computational cost of noise stability measure and training with noise stability regularization relative to average sensitivity or other alternatives? (E.g., the load of extra forward pass on a correlated Y and scaling properties) A brief cost analysis would be useful for practice, but I didn’t see this discussed.

---

> ### Author Response · Authors · 2025-11-16
> **Discussion Comments for Reviewer DkBU**
>
> We thank the reviewer for their engagement and helpful comments which we address below:
> 1. **Comparison with Prior Work:** Placing our work correctly within relevant context is very important and we appreciate the reviewer’s feedback on that aspect:
>     1. We utilize Geometric Influence as a way to generalize average sensitivity towards real functions. This affords us the technical tool required to run an experiment (Figure 1) for analyzing LLM spectra. As a result we find that average sensitivity (and its extension, Geometric Influence) are unsuited to explain the high degree of spectral concentration we observe. This motivates the introduction of noise stability. We will update our manuscript to make sure this point is communicated efficiently.
>     2. The work of Li and Mossel is purely theoretical and assumes a hierarchical structure in the networks involved. They do not analyze Transformers specifically, nor do they contain any empirical results. Rather, examining the noise sensitivity of hierarchical networks they derive space lower bounds for learning problems. We will update our introduction to highlight the distinction between our works in a clearer way.
> 2. **On why the regularizer works and hyperparameter sweeps:**
>     1. This is an important point to clarify and we appreciate the reviewer in bringing it up. Our regularizer encourages stable functions (with wide spectra) because noise stability directly expresses this spectral relationship. Our results in Sections 4 and 5 are meant mainly to reinforce this very point. We will add a reference in our text to clarify this point to the reader.
>     2. We did perform hyperparameter sweeps for the synthetic-tasks, but felt that they did not contribute much insight into the structure and function of our regularizer. As a rule of thumb, we found that $\rho$ should be fairly large in absolute value $(|\rho| \in [0.25,0.75])$ as very small or very large pertrubations intuitively hurt generalization. To further inspire confidence in our results, we run hyperparameter sweeps for $\rho,\gamma$ in the non-synthetic experiments described below. We added this discussion in Section 6.1 of our manuscript
> 3. **On non-synthetic experiments.**
>     1. We thank the reviewer for raising this point. We worked on synthetic datasets because they best showcase the emergence of grokking within our constrained compute environment. However, as per the suggestion of other reviewers as well, we trained atransformer on the next token prediction task for the Wiki-Text-2 dataset. Our findings support the conclusions we drew from our synthetic experiments and updated our manuscript to include them (see updated Section 6.1)
>     * Under noise stability regularization the validation loss and perplexity diminish at the grokking stage much faster than without regularization.
>     * The regularized model achieves higher accuracy than the non-regularized one while only taking 25% as many iterations.
>     * While some choices of $\rho,\gamma$ are better than others (Figure 6), any setting we tried leads to an improvement.
>     * Training with noise stability regularization leads to more stable model, whereas grokking without regularization causes the stability to decrease.
>
> 4. On $S=0$. We added this option because we wanted to stress that stability can be both encouraged and discouraged by our regularizer. We tested $S=1$ only because it is most related to our original motivation of obtaining more stable models. The setting $S=0$ does indeed lead to less stable models, though we have not explored how it can effect realistic language models. We hope that the ability to tune the sensitivity of a model will be a helpful line of future work.
> 5. **Presentation Issues:** we will address typos and small issues in our revision.
> 6. **On Stability Evolution with Regularization:** We thank the reviewer for pointing this out. In our WikiText experiment we do plot noise stability as it evolves with time for regularized vs non-regularized models. This is indeed an insightful plot because it shows that regularized training leads to stable models while non-regularized training leads to unstable models (Figure 6)
> 7. **On the computational cost of noise stability regularization:** This is also a very good and important point and we thank the reviewer for bringing it up. Computing noise stability is no harder than average sensitivity: it takes a constant number of additional forward passes through the network. While not trivial, this is a very scalable cost that could be integrated into training given the benefits of our regularizer. Developing more cost efficient ways to estimate noise stability in a network is a fascinating open problem. We will add a brief discussion of this topic in our manuscript.
>
> Again, we appreciate the reviewer’s help and time. Please let us know if you have any additional questions.

---

> > ### Comment · Reviewer_DkBU · 2025-11-27
> >
> > The authors’ response has addressed most of my concerns. I am now leaning toward acceptance of the paper.

---

### Official Review · Reviewer_vNH8 · 2025-10-31

**Soundness:** 3
**Presentation:** 3
**Contribution:** 3
**Rating:** 6
**Confidence:** 3

**Summary:**

The paper studies noise stability as a quantitative lens on simplicity bias and spectral concentration in Transformers. It formalizes Gaussian noise stability using the Ornstein Uhlenbeck semigroup and Hermite expansions, then proves new results for core components: a closed form for the stability of a ReLU MLP layer, and stability characterizations for attention under identity, low rank, and unstructured regimes. It extends the analysis to depth via a recurrence for MLPs that predicts weak dampening and introduces stability interval propagation to bound stability through deep Transformers. Empirically, the authors propose a noise stability regularizer that is differentiable and data dependent. On grokking style tasks such as modular addition and noisy k sparse parity, the regularizer speeds up emergence by about thirty five percent. They also compare stability based spectral tail bounds with sensitivity based bounds on several pretrained models, finding better agreement with measured concentration when using stability.

**Strengths:**

Stability analysis of attention with identity, low rank, and random regimes is new and insightful. The appearance of a permutation consistency factor $s(\rho)$ in the unstructured case is a neat and interpretable result. Theoretical results are derived from a solid OU semigroup foundation with explicit ReLU formulas and careful discussion of approximations. Interval propagation addresses the realistic case where exact recurrences are brittle. The regularizer is minimal and differentiable, with a clear sampling scheme for $Y$. The experiments control for seeds and hyperparameters and demonstrate a consistent thirty five percent speedup on two canonical grokking tasks. Finally, stability based tail bounds better match measured concentration across common pretrained models than sensitivity based bounds. This supports the relevance of the stability viewpoint for understanding learned spectra in practice.

**Weaknesses:**

- Assumption gaps for deep Transformers. The multi layer recurrence assumes fixed distributions and idealized conditions. The authors later observe full dampening in practice when $\gamma < 1$, which highlights a gap between proxy analysis and actual behavior. More systematic experiments that probe when interval bounds are tight would strengthen claims.

- Limited task scope. The regularizer is only evaluated on synthetic algorithmic tasks. Evidence that it helps on language modeling or fine tuning tasks is indirect through tail bound comparisons. A small scale LM experiment that tracks validation perplexity, stability, and tail mass together would improve external validity.

- Sensitivity to $\rho$ and $\gamma$. The analysis and the regularizer hinge on these hyperparameters. The paper sets them per task but does not map performance and stability as a function of $\rho$ and the regularization strength $\gamma$ across a grid. This makes practical tuning less clear.

**Questions:**

1. Role of  $s(\rho)$ beyond initialization. The unstructured attention result introduces a permutation consistency factor $s(\rho)$ that is derived under random weights. How does $s(\rho)$ evolve during training when attention becomes more structured. Can the authors estimate or bound $s(\rho)$ on trained layers.

2. Tightness of interval bounds. The interval propagation method provides upper and lower bounds. On realistic trained networks, how tight are these intervals per layer. A plot of measured stability with interval envelopes would help assess utility.

3. The regularizer allows both orientations. Have you tried S=0 to discourage stability as a stress test on overfitting or memorization. Does decreasing stability hurt grokking or robustness.

4. From synthetic to text. Can you run a small language modeling experiment that logs stability, spectral tail, and validation perplexity every few thousand steps. Even a 100M parameter model on WikiText or FineWeb subsets would help connect theory to practice.

---

> ### Author Response · Authors · 2025-11-16
> **Discussion Comments for Reviewer vNH8**
>
> We thank the reviewer for their insightful and thorough review. We address their concerns and questions below:
>
> 1. **On the distributional assumptions we make about multi-layered Transformers**:
>     1. We agree with the reviewer that analyzing multi-layered networks under idealized conditions is not the most desirable setting. Removing these assumptions is a major open question in the field of moment propagation in neural networks, and it is standard for prior work to make such idealized assumptions. While we are unfortunately unable to remove such assumptions ourselves, our analysis in Section 5.3 and Appendix H does show that self-attention alters the distribution enough that standard recurrence-based analyses are perhaps too naive. This is a new insight in itself that we hope can motivate further investigation into explaining this gap between proxy analysis and practice.
>     2. We agree that more systematic experiments probing when interval bounds are tight would help strengthen our claims. We proposed interval propagation mainly as a theoretical alternative to analytic computation of moment propagation, and so we initially felt such experiments exceed the scope of our work and decided to leave them as a future direction. We updated our manuscript to reflect the necessity of exploring this direction further.
> 2. **Evaluating the regularizer with larger LMs:** We thank the reviewer for suggesting this new direction to us. We agree that evaluation on non-synthetic datasets would be very helpful towards studying our regularizer. As per the reviewer’s suggestion, *we trained our transformer model (around 150K total parameters) on WikiText-2 (Small) for the Next Token Prediction task and examined noise stability and the effect of the regularizer in this setting*. We also performed a hyperparameter sweep of the $\gamma$ and $\rho$ parameters. Our findings support the conclusions we drew from our synthetic experiments and updated our manuscript to include them (*see updated Section 6.1*)
>     * Under noise stability regularization the validation loss and perplexity diminish at the grokking stage much faster than without regularization.
>     * The regularized model achieves higher accuracy than the non-regularized one while only taking 25% as many iterations.
>     * While some choices of $\gamma$ and $\rho$ are better than others (*Figure 6*), any setting we tried leads to an improvement.
>     * Training with noise stability regularization leads to more stable model, whereas grokking without regularization causes the stability to decrease.
>
> 3. **Sensitivity to $\rho$ and $\gamma$:** Our synthetic experimental findings are a result of a parameter sweep for $\rho$ and $\gamma$, which gave different optimal outcomes for different tasks. While we cannot provide a specific rule for setting them, we empirically observed that $\rho \in [0.25,0.75]$ is often crucial, which can intuitively be explained by the fact that very heavy or very light perturbations to the input do not help generalization. As for $\gamma$, we empirically observed it correlates with the sensitivity of the function: low sensitivity functions like sparse parity require a smaller intervention from the regularizer and thus a smaller $\gamma$. We also give a more thorough picture of this landscape in our new experiments.
>
> Questions:
>
> 1. **On** $s(\rho)$: The reviewer raises an interesting question. When the weights are randomly initialized, we showed that the attention matrix consists of Dirac-like spikes for every row. Under random perturbations, we showed that these spikes permute according to some $s(\rho)$ function. An analogous definition for general weights would have to take into account that this spikes may no longer be present. This regime is of interest because in LMs the attention weights are typically very concentrated. Studying this question is definitely a compelling future direction!
> 2. **Tightness on interval bounds:** As we discuss above, we agree that giving empirical backing to our interval bounds is an important future direction that will give more substance to our theoretical method. Interval propagation is, in the context of our paper, a theoretical insight that we hope can be helpful in analyzing moment propagation in deep networks.
> 3. $S=0$: This setting does indeed lead to less stable models, though we have not explored how it can effect realistic language models. It is likely that it encourages memorization, as the reviewer suggests, though this is an experiment we leave for future work.
>    * We tested $S=0$ for the Wiki-Text experiment and noticed that it leads to high stable, but poorly performing models.
> 4. **Synthetic → Text Experiment:** See discussion above.
>
> We thank the reviewer again for engaging and their insightful comments. We appreciate their time and help. Please let us know if there is any other questions you may have.

---

> > ### Comment · Reviewer_vNH8 · 2025-11-22
> >
> > Thank you for your response and the new experiments on wikiText-2 really demonstrated the effectiveness of the proposed regularization. I'll continue supporting this paper.

---

### Author Response · Authors · 2025-12-01

We thank all the reviewers and area chairs for their time and engagement with our work. We wanted to highlight how our response has addressed the main concerns voiced during the rebuttal phase:
1. **Experiments on non-synthetic tasks**: As many reviewers noted, including non-synthetic experiments would strengthen the case for using our proposed regularizer. In response, we conducted an additional experiment on the wikitext-v2 dataset for the task of character prediction and tested noise stability regularization. We indeed found that our method leads to validation accuracy and loss dropping much faster than without using regularization, further supporting our claims. We included these experiments in Section 6.1 and Appendix H.3 of our manuscript.
2. **Ablation studies**: we included in our appendix further ablation studies testing different values of $\rho$ and $\gamma$ for our regularizers.
3. **On average sensitivity regularization**: We compared our regularizer with average sensitivity regularization by running another experiment. Though we left a full exploration of the effects of these regularizers for future work, we did identify a catalyzing effect in average sensitivity regularization as well, for the non-synthetic experiments. This effect is less pronounced than in our regularizer, but very interesting nonetheless.

We hope that via this discussion most concerns and points of confusion have been addressed. Thank you again for your insightful comments.

---

### Meta-Review · Area_Chair_QcA8 · 2026-01-03

**Summary:**

The authors study the notion of noise stability as a measure of simplicity bias and the effect of regularization on the noise stability. Most reviewers viewed this submission positively, seeing the noise stability as a useful metric to analyze transformers in the future. While the main critique seems to be mostly focused on the lack of comparison against previous work on geometric influence and other regularizers.

I'm somewhat disappointed as no one drew the connection between the notion of noise stability and signal propagation, which has now a very extensive line of work, and understanding the impact of the propagation and training dynamics, including asymptotic characterizations. See the following two papers for example:
[1] https://arxiv.org/abs/2110.01765
[2] https://arxiv.org/abs/2405.15712

To be precise, the noise stability measure treats two inputs $X, Y$ as correlated Gaussians (related to NNGP but not required to be Gaussian), and measures what effectively is the covariance of the outputs $\mathbb{E} f(X) f(Y)$ after a layer $f$.

This is exactly how one would study the feature kernel propagation across layers! At initialization, this is exactly what [1] refers to as $C$ maps, and during training, it's characterized as $H$ kernels in [2], which can even be taken to infinite depth limits with residual connections. It is now well understood that the $H$ kernel plays a significant role in the training dynamics, as it directly shows up in the DMFT equations for all architectures (there are also other kernels that play equally significant roles).

I should also emphasize that, the whole point of [1] is to demonstrate that well regulated signal propagation leads to both stable propagation (i.e. noise stability will not dampen to zero) and significantly faster training dynamics! Which means the accelerated results on grokking is really not a surprise at all. There have been also many follow up works towards this direction, where proper signal propagation yields significant improvements in training.

So this puts me at a crossroad. On one hand, clearly the authors were on the right track, as the notion of noise stability is definitely important, and have received overall fairly positive reviews. On the other hand, the authors are accidentally rediscovering an important concept, which has now been very studied, with the literature quite a bit further ahead. At the same time, none of the reviewers have noticed this connection to an entire line of literature makes my criticism feel like it's coming too late without giving the authors a chance of respond. This whole ICLR process of reassigning ACs and makes me want to lean towards the forgiving side.

Therefore, I ultimately decided that the paper should be accepted conditional on the authors providing a thorough discussion on the connection with the line of work on signal propagation, and especially the relationship with recent work on infinite width/depth limits of transformers.

**Reviewer Concerns:**

Addressed:
vNH8: synthetic data, hyperparameter sensitivity, and orientation stress test.
DKBU, 6UoK: comparison with geometric influence, regularizer intuition.

Unaddressed:
vNH8: noise stability dampen to zero (albeit solution is known from other works).
QkLJ: comparison with other regularizers, tighter tail bounds, limited experiments.

**Reviewer Scores:**

6UoK - 8
vNH8 - 6 (potential raise to 7)
DkBU - 4 (likely raise to 5)
QkLJ - 2

---

### Decision · Program_Chairs · 2026-01-26

Accept (Poster)